# Comparative Analysis of Volatile Flavor Compounds in Strongly Flavored Baijiu under Two Different Pit Cap Sealing Processes

**DOI:** 10.3390/foods12132579

**Published:** 2023-07-01

**Authors:** Lingshan Li, Mei Fan, Yan Xu, Liang Zhang, Yu Qian, Yongqing Tang, Jinsong Li, Jinsong Zhao, Siqi Yuan, Jun Liu

**Affiliations:** 1Bioengineering College, Sichuan University of Science & Engineering, Yibin 644000, China; lilingshan@stu.suse.edu.cn (L.L.); fanmei@stu.suse.edu.cn (M.F.); 2School of Biotechnology, Jiangnan University, Wuxi 214122, China; yxu@jiangnan.edu.cn; 3Luzhou Laojiao Group Co., Ltd., Luzhou 646000, China; zhangl@lzlj.com (L.Z.);; 4Analysis and Testing Center, Sichuan University of Science & Engineering, Zigong 643000, China; qianyu1001@sina.com; 5Sichuan Liquor Group, Luzhou Tianfu 1st Street (Liangjiang International), Wuhou District, Chengdu 610000, China; zhaojinsong@suse.edu.cn; 6Science and Technology Department, Sichuan University of Science & Engineering, Zigong 643000, China; 7Key Laboratory of Liquor-Making and Application, Sichuan University of Science & Engineering, Yibin 644000, China

**Keywords:** flavor compounds, strong flavor Baijiu, pit cap, mud-sealing pits, steel-sealing pits, Gas Chromatography-Mass Spectrometry

## Abstract

The solid-state fermentation process of strongly flavored Baijiu is complicated by the co-fermentation of many different microorganisms in the fermentation pools. The traditional fermentation pools of strong flavor Baijiu are sealed with mud, and this sealed-pit mud is not easy to maintain; therefore, the pit cap is prone to cracks and to caving in. The destruction of the sealed-pit mud may lead to instability in the composition and an abundance of microorganisms in the fermentation process that results in fluctuations of product quality. Thus, the production method of replacing the mud cap with a new steel cap is gradually attracting the attention of scientific and technical workers in the industry. However, so far, there have been relatively few reports on the use of steel lids for sealing pits for fermentation and brewing. In this study, the volatile flavor components of 270 Baijiu samples from mud-sealing and steel-sealing pits of a Chinese Baijiu distillery were studied qualitatively and quantitatively using Gas Chromatography–Mass Spectrometry (Abbreviated as GC-MS). Our statistical methods included Hierarchical Cluster Analysis (Abbreviated as HCA), Principal Component Analysis (Abbreviated as PCA), and Discriminant Analysis (Abbreviated as DA). A statistical analysis was carried out on the yield of strongly flavored Baijiu, and we made a comprehensive evaluation of the Baijiu produced under the two pit-sealing modes with regard to flavor and economic efficiency. The yield of strong flavored Baijiu was 6.7% higher with steel-sealing pits compared with mud-sealing pits. Cluster analysis categorized the strongly flavored Baijiu samples into two categories initially: (1) samples produced using mud-sealing pits and (2) samples using steel-sealing pits. Our analysis also indicated that the 28 compounds used for quantification were selected correctly. Surprising to the experimental staff, the overall score for the steel-sealing pits was greater than that of the mud-sealing pits based on PCA. Using DA, the prediction results were 100% accurate. In summary, through a comparative analysis of the flavor and yield, which are the two main factors that affect the quality of Baijiu in a distillery, and systematic combination at both experimental and theoretical levels, the differences between the Baijiu production by steel-sealing and the traditional mud-sealing were clear. Regardless of the impact of age, the detectable flavor components of Baijiu from the mud-steeling pits were very consistent with those of the steel-sealing pits in terms of richness or concentration. However, steel-sealing pits were significantly superior to mud-sealing pits with respect to output, consistency in quality, and cost (human and economic) savings.

## 1. Introduction

By the end of 2022, the cumulative export value of alcoholic beverages in China was approximately 21.53 billion RMB, an increase of 25.1% year-on-year (China Industrial Economic Information Network). Topping the list of alcoholic beverages in China was Chinese Baijiu, which includes strong flavor, Maotai flavor, light flavor, and others [1,2,3,4]. The unique brewing process, which involves mixed steaming and boiling, and continuous ingredients for fermented grains of Chinese Baijiu endow itself with characteristic flavor [3]. Chinese Baijiu is highly sought after by consumers and is also renowned overseas. The fermentation of Baijiu is the result of the joint action of microbial clusters. Good fermentation technology will lead to the production of a large number of flavor substances and metabolites, which not only have rich nutritional value but also have certain medicinal value, such as cyclic dipeptide, prochemic acid, etc. [5,6].

Strongly flavored Baijiu is produced from grains with medium-high temperature, with Daqu being the flavor-producing agent, and involves continuous distillation of grain ingredients, mixed steaming and mixed burning, solid-state fermentation, distillation, aging, and blending [7]. Aromatic compounds in Baijiu mainly come from protein or amino acid metabolism. It has a strong aromatic odor and has a significant fragrance forming effect [8,9,10]. In fact, the aromatic compounds in Baijiu come from fermentation during the brewing process and decomposition or synthesis during distillation. Among them, esters, aldehydes, and ketones produced during fermentation have aromatic properties and are the main source of aromatic compounds in Baijiu. In the distillation process, with the evaporation of alcohol, some volatile aromatic compounds will also be distilled out and become the aromatic substances in Baijiu. Their aromatic compounds are dominated by ethyl caproate and are produced without the addition of edible alcohols, nonfermented aromatic compounds, or taste-producing substances [11,12]. The fermentation of strongly flavored Baijiu is carried out in mud pits (usually called “NiJiao” in Chinese) [13], which have an important influence on the formation of flavor compounds in strongly flavored Baijiu. With the increase in demand for Baijiu, the number and scale of Baijiu enterprises has expanded. In the face of growing market demand, traditional manual production is unable to meet the production demands for Baijiu products; the intelligent production and automation in distilleries is gradually being put on the agenda by researchers in the industry. Traditional strongly flavored Baijiu is brewed by sealing the pit with mud [14] (Figure 1). This process isolates oxygen in the early stage of production and prevents the microbial flora in the environment from invading the fermentation system. However, as fermentation time passes, this method of sealing the pits has shortcomings, such as cracks in the pit cap, maintenance of the pits, and contamination with environmental bacteria [15,16]. These factors affect the quality and taste of strongly flavored Baijiu, which causes some unnecessary, but significant, economic losses.

The sealing technology of pit is one of the key points in the production of Chinese Baijiu possessing varied and diversified flavors (12 types including Maotai flavor, strong flavor, Qingxiang flavor, etc.). The key fermentation elements concerned such as the types of fermentation microorganisms and the corresponding fermentation time are all related to the sealing of the pit. Poor sealing technology can lead to a series of devastating consequences such as fermentation rancidity, heavy odor, and low alcohol production rate. However, better sealing technology can make the aroma of the Baijiu stronger and more unique.

In response to these problems, different forms of pit sealing have been developed by Baijiu practitioners. For example, steel-sealing methods have been improved from the simple steel-sealing lid (Figure 2) to those that are detectable from the outside (Figure 3) [17], and plastic film sealing has also started to be applied in the fermentation sealing process of strongly flavored Baijiu. The plastic film sealing pit has disadvantages, such as difficult operation, poor insulation, and easy breakage. On the other hand, the steel-sealing pit solves the problems of entering and exiting the traditional pit fermented grains, high labor intensity, poor control of the internal temperature and humidity of the pit, and the problems of automating the production of the solid fermentation type of strongly flavored Baijiu. It also allows real-time monitoring of the physical and chemical indicators related to fermentation in the pit. Therefore, the production method of replacing the mud-sealing cap with a new steel-sealing cap has gradually received the attention of scientific and technical workers in the industry, but relatively little research has been reported on the fermentation and brewing method of steel-sealing caps.

The quality of Baijiu products is closely related to the composition of flavor substances. For Chinese Baijiu with large export volume, the method of steel-sealing pits can be dynamically monitored from the outside, which can not only reduce a series of problems caused by mud-sealing pits but also reduce the production of harmful substances through monitoring. At present, the research on the flavor substances of strongly flavored Baijiu mainly focuses on volatile substances, which play an important role in Baijiu aroma. Compared with other analytical techniques, Gas Chromatography–Mass Spectrometry (GC-MS) has obvious advantages in flavor analysis [18,19,20,21,22,23,24,25]. GC-MS can be used to analyze qualitatively and quantitatively most of the volatile flavor components in strongly flavored Baijiu; for this reason, GC-MS has become one of the most powerful tools for the separation and detection of the larger number of flavor compounds in Baijiu [26,27].

To identify the differences between the quality of Baijiu products produced in mud-sealing and steel-sealing pits systematically, we collected strongly flavored Baijiu samples from mud-sealing and steel-sealing pits from a distillery using GC-MS to analyze flavor, studied the volatile flavor components using Hierarchical Cluster Analysis (Abbreviated as HCA), Principal Component Analysis (Abbreviated as PCA), and Discriminant Analysis (Abbreviated as DA), and analyzed the results statistically. The ultimate goal was to find a way to achieve both artificial intelligence and intensive production to save time, effort, and money. We hoped to determine whether the quality and taste of the product (Baijiu) was affected, compared with the traditional method of production, to solve the increasingly serious challenge between supply and demand faced by the Baijiu industry.

## 2. Materials and Methods

### 2.1. Materials

Materials: Baijiu samples were taken from the same vintage and same fermentation pits at a famous Baijiu enterprise in southern Sichuan. Samples were collected from steel-sealing and traditional mud-sealing pits.

Distillation: For ease of operation, this facility distilled Baijiu in upper, middle, and lower layers of grain.

Strongly flavored Baijiu picking: Because Chinese traditional Baijiu uses solid-state, retort barrel distillation technology, the quality and alcohol concentration of distilled Baijiu from different time periods are different, and the Baijiu is usually picked and stored separately in stages during the production process. In this study, the distilled strongly flavored Baijiu (i.e., first segment Baijiu, second segment Baijiu, and third segment Baijiu) in the corresponding Baijiu period was picked by a Baijiu picker.

Sample collection time: The brewing workshop for the strong flavor Baijiu distillery was divided into five times every year: the first time was in September; the second time was in November; the third time was in February of the next year; the fourth time was in April of the next year; and the fifth time was in June of the next year.

### 2.2. Instruments and Equipment

We used a TSQ8000 triple quadrupole GC-MS, an AI1310 multifunctional autosampler (Thermo Fisher Scientific, Waltham, MA, USA), a pipette gun (1000 μL, Dalong Xingchuang Experimental (Instruments) Co., Ltd., Beijing, China), a volumetric flask (100 mL), an injection bottle (2 mL), a TG-WAX column (30 m × 0.25 mm × 0.25 μm, Aglient Corporation, Santa Clara, CA, USA), an ultrasonic cleaner, and a microinjector 1 μL.

### 2.3. Chemicals

Flavor substance standards were chromatographic grade or above. *n*-propanol, ethyl valerate, *n*-butanol, isobutanol, acetic acid, hexyl butyrate, isoamyl alcohol, *n*-amyl alcohol, ethyl butyrate, *n*-hexanol, hexanoic acid, ethyl formate, ethyl acetate, ethyl caproate, ethyl heptanoate, butyl caproate, ethyl octanoate, propionic acid, butanoic acid, heptanoic acid, octanoic acid, and *tert*-amyl alcohol were purchased from Tianjin Guangfu Chemical Reagent Co., Tianjin, China. Isoamyl acetate, *n*-amyl acetate, and anhydrous ethanol were purchased from Beijing Tanmo Quality Control Standard Material Centre., Beijing, China. Ethyl phenylacetate, ethyl tetradecanoate, and ethyl hexadecanoate were purchased from Bailingway Technology Co., Ltd., Beijing, China. Propyl hexanoate and ethyl linoleate were purchased from Aladdin Reagent (Shanghai) Co., Ltd., Shanghai, China. Anhydrous ethanol was purchased from Shanghai Aladdin Biochemical Technology Co., Ltd., Shanghai, China. Methanol was purchased from Adamas-beta Co., Shanghai, China. 

### 2.4. Methods

#### 2.4.1. Calculations of Yield of Strongly Flavored Baijiu 

Yield rate = brewing strong flavor Baijiu yield/amount of grain × 100%. The mud-sealing pit contained 130 kg of grain per retort, and the steel-sealing pit contained 225 kg of grain per retort.

#### 2.4.2. Identification of Volatile Flavor Compounds Using GC-MS

Qualitative analysis was carried out by combining the retention index (RI) of the flavor substances with a NIST 12 mass spectrometry library. The method for the quantitative analysis of flavor components in strong flavor Baijiu was based on the national standard GB/T 10345-2007 [28] “Methods for the analysis of Baijiu”. The specific processing steps of the sample were as follows: We measured the standard substance of flavor components and used 60% (*v*/*v*) ethanol solution to volume with a 100 mL volumetric flask. The measured value of all flavor components was 0.02% *v*/*v*. Before the samples were analyzed, the mixed standard solutions were analyzed, and the correction factors for each flavor compound were calculated using the peak areas and the concentration of the standard substances. Amyl acetate, 2-ethylbutyrate, and tert-pentyl alcohol were used as internal standards for esters, acids, and alcohols of flavor components, respectively.

#### 2.4.3. Statistical Analysis 

HCA, PCA, and DA were applied to extract and to analyze the data. Wherein, the specific methods for PCA are as follows:

The formula for each principal component score is Fi = w_i1×1_ + w_i2_X_2_ + … + w_in_X_n_

Among them, $W_{ij}=\theta_{i}/\sqrt{\lambda }$, which denotes the weight of each variable in the principal component. $\theta_{i}$ is the coefficient that corresponds to each variable in the component matrix, and $\sqrt{\lambda }$ is the open root value of the eigenvalue that corresponds to the i-th principal component.

We calculated the comprehensive score by using the contribution rate of the Baijiu on seven main components in different pit sealing methods as the weight coefficient of the comprehensive score; the formula was F = α_1_F1 + α_2_F2 + … + α_n_Fn, where α_i_ denotes the percentage of variance of the ith principal component.

ChemPattern software V2017.3 (Comayne Beijing Technology Co., Ltd., Beijing, China) and SPSS 24.0 were also used to analyze the data chemometrically. Excel 2016 and Origin 2018 software were used to collate statistics and to draw graphs for analysis of the assay data.

## 3. Results and Discussion

### 3.1. Sample Collection

According to the principle of parallel sampling, three pits were selected for mud and steel sealing, and samples were taken from each of the three pits selected. We sampled the upper, middle, and lower layers of fermented grains from each pit, and the 1st, 2nd, and 3rd segment samples of strong flavor Baijiu were obtained from distillation of each layer of fermented grains, for a total of 270 samples. For example, we use mud-sealing pits for sampling: 3 pits × 3 (upper, middle, and lower layers of fermented grains) × 3 (each layer has 1st, 2nd, and 3rd segment) × 5 (a total of five samples were taken) = 135.

### 3.2. Yield of Strongly Flavored Baijiu

The yield in strongly flavored Baijiu from the same pit at different sampling times showed that the yield decreased from spring to autumn to winter to summer when we used either traditional or new sealed pits (Figure 4). The steel-sealing pit cap exhibited less variation in yield than the mud-sealing cap, which provided preliminary evidence of the stability of sealing with steel. In a fast-moving society, the more stable the Baijiu yield is for roughly the same quality of Baijiu, the better this is. This result indicated that the environment had little influence on the outcome. The greater stability of sealing with steel should improve the economy of the distillery operation, and this provides more theoretical relevant data for the intelligent operation of the distillery.

The yield from the two different forms of pit sealing (Figure 5) showed that for the first segment of Baijiu, the upper, middle, and lower lees yielded 20.00–24.10% (average 21.31%) in the mud-sealing pit, while the steel-sealing pit yielded 19.07–24.3% (average 21.61%). For the second segment, yield was 19.90–21.74% (average 20.72%) at each level of the lees in the mud-sealing pits, while 20.22–29.27% (average 24.31%) was produced in the steel-sealing pits. For the third segment, the yield of each layer of lees in the mud-sealed pit was 9.70–14.31% (average 11.58%), while the steel-sealing pit yielded 8.76–11.81% (average 10.04%). For the pits as a whole, the rate of strongly flavored Baijiu yield from the upper, middle, and lower lees in the mud-sealing pits was 37.03–45.21% (average 41.42%), while the rate of strong flavor Baijiu yield in the steel-sealing pits was 44.35–51.23% (average 48.12%). 

The second segment Baijiu was characterized by its high alcohol concentration and high ester concentration, and their aromas were rich, pure, and harmonious [29]. In addition, the quality of the second segment Baijiu was excellent, and for the corresponding pits with a small difference between the first segment Baijiu (0.30% difference between mud-sealing and steel-sealing pits) and the third segment Baijiu (1.54% difference), the yield of the second segment Baijiu in the steel-sealing pits (24.31%) increased compared with the second segment in the mud-sealing pits (20.72%). Overall, Baijiu quality improved.

From a comprehensive point of view, the rate of strongly flavored Baijiu production in steel-sealing pits increased in the upper and middle tiers of lees. The application of steel lids improved both quality and production of Baijiu significantly.

### 3.3. Analysis of Volatile Flavor Components in Strongly Flavored Baijiu Produced under Two Types of Pit Sealing Methods

#### 3.3.1. Qualitative Analysis

The 270 strongly flavored Baijiu samples were studied qualitatively and quantitatively using GC-MS with a direct injection method. A total of 112 compounds were detected within 32 min, which included fifty-eight esters, fourteen acids, twenty-two alcohols, six aldehydes, seven ketones, and other compounds (Table 1). The analysis of the flavor compounds in the steel-sealing and the mud-sealing pits showed that there were 104 volatile flavor compounds in the mud-sealing pit and 102 volatile flavor compounds in the steel-sealing pit.

Further collation of the common and characteristic volatile compounds in the upper, middle, and lower lees in mud and steel sealing pits (Table 1) showed that the types of volatile compounds in the strongly flavored Baijiu of the mud cap pit increased from top to bottom with the fermented grain layers, and the lower layer of fermented grains had more types of compounds (Figure 6). The lower layer of fermented grains was in contact with the pit wall and the bottom of the fermentation pit. The surface of the mud pit of the cellar had more microorganisms, and more metabolites were produced in this complex microbial system. In addition, the lower layer of fermented grains was immersed under the yellow water line, and the composition of the yellow water was very complex. This composition consisted mainly of starch, polysaccharides, dextrins, proteins, some minerals, alcohols, aldehydes, acids, and esters, which can provide reasonable nutritional and material conditions for the growth of microorganisms, thus, causing the lower layer of grains to contain more flavor substances. There was no significant difference in the types and quantities of flavor compounds from different layers of fermented grains in the steel-sealing pit, probably due to the steel-sealing method, which allowed controlled internal fermentation that we observed through external monitoring. 

In addition, the following volatile compounds with antibacterial, analgesic, and anti-inflammatory activity were found in the middle fermented layer of grains in the steel-sealing pit: L-pyroglutamic acid methyl ester, and Cyclo (Phe-Pro) [30]. These compounds are beneficial when consumed and provide a basis for the direction of subsequent research on beneficial flavor compounds in this brand of Chinese Baijiu.

#### 3.3.2. Further Analysis of Esters, Acids, and Alcohols in the Volatile Flavor Compounds

Esters are the main aromatic substances in strongly flavored Baijiu, with a pleasant fruity aroma, and they were dominated by ethyl caproate, ethyl acetate, ethyl lactate, and ethyl butyrate in our samples [31,32]. Esters are important substances that affect the flavor of Baijiu. Esters included ethyl acetate, ethyl caproate, ethyl lactate, ethyl butyrate, ethyl palmitate, ethyl anti oleate, ethyl valerate, ethyl heptanoate, and ethyl caproate in descending order of percentage, which conforms to the basic characteristics of strongly flavored Baijiu. The area percentage of ethyl acetate was the highest (3.66–18.53%), followed by ethyl caproate (4.34–16.57%), ethyl lactate (1.52–8.79%), ethyl butyrate (0.59–4.65%), ethyl palmitate (0.29–7.07%), ethyl trans oleate (0.03–5.73%), ethyl valerate (0.25–1.57%), ethyl heptanoate (0.11–0.96%), and ethyl octanoate (0.10–1.26%).

The following four types of alcohols were detected in all samples and accounted for a large proportion of all alcohols in this test: ethanol, n-butanol, propanol, and isobutanol. As the main component of Baijiu, ethanol concentration has an important impact on the yield of Baijiu. The following acids were detected at high levels: acetic acid, caproic acid, and butyric acid. Lactic acid, which is the precursor of ethyl lactate, was detected in a small amount in this test. It may be related to the weak volatility of lactic acid through analysis. In a future study, appropriate pretreatment methods can be selected to treat the samples to enrich the substances with weak volatility to obtain a more comprehensive detection of the flavor compounds of Baijiu.

#### 3.3.3. Quantitative Analysis

Compared with the samples with more quantitative substances and considering that the Chinese Baijiu distillery uses single grain brewing, the volatile flavor components from single grain brewing contained fewer total esters compared with multi-grain brewing. We screened a total of 28 flavor components with higher concentration for quantitative analysis after determining the ethanol peaks; a total of 28 flavor components with a peak height >0.5% were selected for quantitative analysis, of which sixteen were esters, six were alcohols, and six were acids. The concentrations of flavor compounds in the steel-sealing pits were compared with the corresponding stratified (upper, middle, and lower) and segmented (1st, 2nd, and 3rd segments) pits in mud-sealing pits (Table 2, Table 3 and Table 4).

Alcohols are produced under aerobic conditions through the deamination of amino acids or the decarboxylation of sugars [33,34,35], most of which display a distinctive fruity flavor. In this study, six alcohols, which included n-propanol, isobutanol, n-butanol, isoamyl alcohol, n-amyl alcohol, and n-hexanol, were analyzed quantitatively in all samples. The quantitative situation is detailed in Table 2, Table 3 and Table 4.

Organic acids are the important flavor substances of strongly flavored Baijiu, and they are also the precursors of the flavor components of Baijiu. An appropriate amount of organic acids can make the Baijiu plump, harmonious, and with a long aftertaste. Among them, caproic acid, butyric acid, and acetic acid are the chromatographic skeletal components, which have obvious flavor-fixing effects in Baijiu. The corresponding esters generated by the interaction of different kinds of organic acids and alcohols constitute the main flavor substances of strongly flavored Baijiu [34]. When the concentration of these esters in all Baijiu samples was low, they exhibited a certain cheesy aroma, which increased the complexity of the aroma of strong flavor Baijiu.

**Table 1 foods-12-02579-t001:** Identification of flavor components of strongly flavored Baijiu under two pit sealing methods.

No.	Compounds Specific to Mud-Sealing Pits, Steel-Sealing Pits		Volatile Flavor Compounds	Descriptions [36]
1			Acetaldehyde	Pungent, ether-like odor, fruity, coffee, wine and green aromas when diluted
2			2-Propanone	Pungent, sweet and slightly aromatic
3			Ethyl methanoate	Fruity aroma
4			Ethyl acetate	Pineapple scent
5			Methanol	Pungent odor
6			2-Methylbutyraldehyde	Asphyxiating odor
7			Isovaleraldehyde	Aroma of apple at high dilution, peach at concentration below 10 ppm
8			2-Hydroxy-3-pentanone	
9			Ethyl 2-methylpropanoate	
10			Propane,1,1-diethoxy-2-methyl-	
11			1,1-Diethoxy-pentan	
12			2-Butanol	The smell of wine
13			Ethyl butyrate	Sweet and fruity, with notes of pineapple, banana and apple
14			1-Propanol	Strong aromas of meat at low concentrations
15			Acetaldehyde butyl ethyl acetal	
16			Ethyl isovalerate	Apple, mulberry aroma
17			Isovaleraldehyde diethyl acetal	
18			Butyl acetate	Fruity aroma, diluted with a pineapple, banana-like aroma
19			Isobutanol	Alcoholic, irritating odor
20			Isoamyl acetate	
21			2-Butanol, 3-methyl	
22			Ethyl valerate	Fruity, sourness
23			1-Butanol	Jasmine, spicy flavor
24			Amyl acetate	Fruity scent
25			Methyl hexanoate	Volatile, etheric aroma, pineapple-like
26			3-Methyl-1-butanol	Mixed alcoholic and spicy notes with mellow, etheric and banana aromas.
27			Butyl butyrate	Apple scent
28			2-Hexanol	
29			Ethyl caproate	Fruity aroma
30			1-Pentanol	
31			Isoamyl isobutyrate	
32			3-Hydroxy-2-butanone	Sweet, dairy aroma with a fatty, oily note
33			Imidazole-4-acetic acid	
34			1,1,3-Triethoxypropane	
35			Pentanoic acid, butyl ester	Sweet fruity, green aromas
36			Caproic acid propyl ester	Elegant aromas of pineapple and blackberry undertones
37			(S)-(+)-2-Heptanol	
38			Ethyl heptanoate	Fruity, green, waxy, Cornish
39			Ethyl L(-)-lactate	Sweet, tart and fruity aroma
40			1-Hexanol	
41			Methyl 2-hydroxyisobutyrate	
42			Hexanoic acid butyl ester	Elegant aromas of pineapple and apple
43			Hexyl butyrate	Green, waxy, and fruity, with a characteristic aroma of almond fruit
44			3-Methyl-2-butanol	
45			Ethyl caprylate	Waxy aroma, musty aroma, fruity apricot-like aroma, creamy aroma, milk aroma, sweet wine aroma
46			Hexyl acetate	Green, sweet and fruity aromas with hints of appleand banana peel
47			Ethylidene diacetate	
48			1-Hydroxy	Fresh, light, oily aroma with wine notes and a spicy flavor
49			Acetic acid glacial	Irritating odor
50			Furfural	Special scent
51			DL leucine ethyl ester	
52			Butyl lactate	Slightly smelly
53			Ethyl nonanoate	Fruity, waxy, estery and green aromas
54			Isoamyl lactate	
55			Isobutyric acid	Pungent odor
56			(S)-(+)-1,2-Propanediol	
57			Hexyl hexanoate	Green, waxy, herbal and tropical fruit and berry aromas.
58			Octanoic acid, butylester	
59			Propionic acid	Irritating odor
60			Butyric acid	Putrid sour smell
61			Ethyl caprate	Aromas of grapes, Cornish wine
62			Phenylacetaldehyde	Hyacinth aroma
63			Furfuryl alcohol	Special smell and bitter spicy taste
64			Isovaleric acid	Pungent sour odor, with the aroma of cheese, dairy products and fruits after dilution
65			Heptyl formate	Floral and fruity aromas
66			(2,2-Diethoxyethyl)-Benzene	
67			2-Ethylbutyric acid	Sour, musty odor
68			Ethyl phenylacetate	Floral, fruity, powdery, woody, animal, cocoa
69			Hexyl caprylate	Fruity, green, waxy, ester
70			Phenethyl acetate	Rose aroma with dense sweet undertones
71			Ethyl dodecanoate	Gentle fragrance
72			Hexanoic acid	Sweat, cheese, sourness
73			Heptanoic acid	Fermented, waxy and fruity aromas
74			Octanoic acid	Sweat odor
75			Valeric acid	Sweat, cheese, sourness
76			Cyclopentadecanolide	
77			9-Ethyl oxynicotinate	
78			Pentadecanoic acid,ethyl ester	
79			2-Pentadecanone,6,10,14-trimethyl	
80			9-Hexadecenoic acid,ethyl ester	
81			9,9-Diethoxynonanoic acid ethyl ester	
82			Hexadecanoic acid ethyl ester	
83			Ethyl 3-phenylpropionate	
84			Ethyl 12-oxododecanoate	
85			2-Phenylethanol	Rose fragrance
86			9-Hexadecenoic acid	
87			2-[[(9Z,12Z)-9,12-Octadecadienyl]oxy]ethanol	
88			Heptadecanoic acid,ethyl ester	
89			Methyl (7Z)-7-hexadecenoate	
90			Ethyl myristate	
91			Octadecanoic acid,ethyl ester	
92			Ethyl oleate	Floral, fruity and oily aromas
93			Ethyl linoleate	
94			5,8,11,14-Eicosatetraenoicacid	
95			Ethyl alpha-linolenate	
96			Methyl linolenate	
97			Hexadecanoic acid	
98			2-Butanol	
99			2-Heptanol	
100			Heptyl heptanoate	
101			Methyl L-pyroglutamate	
102			3-Isobutyl-2,3,6,7,8,8a-hexahydrOpyrrolopyrazine-1,4-dione	
103			Diethyl succinate	
104			Isobutyl hexanoate	Sweet fruity, green and waxy aromas
105			Heptanol	Fresh, lightly oily aroma with hints of wine and a spicy flavor
106			2-Hexadecanol	
107			2-Thiapropane	
108			Hexadecanoic acid, butyl ester	
109			Cyclo(Phe-Pro)	
110			5-Methyl-2-heptanone	
111			Methyl trans linoleic acid ester	
112			L(+)-Lactic acid	

The compounds in the table are sorted in the order of peak appearance. The colors in the table above each represent the following: Blue: Specific compounds of mud-sealing pit in the upper layer of fermented grains. Green: Specific compounds of steel-sealing pit in the upper layer of fermented grains. Brown: Specific compounds of mud-sealing pit in the middle layer of fermented grains. Grey: Specific compounds of steel-sealing pit in the middle layer of fermented grains. Purple: Specific compounds of mud-sealing pit in the lower layer of fermented grains. Orange: Specific compounds of steel-sealing pit in the lower layer of fermented grains. White: The compounds contained in the upper, middle, and lower fermented grains that corresponded to the mud-sealing pit and the steel-sealing pit.

**Table 2 foods-12-02579-t002:** The concentration of flavor compounds in the upper, middle, and lower layers of the first segment of strongly flavored Baijiu in the pit under two pit sealing methods.

Flavor Substances	Flavor Substance Concentration/(mg·L^−1^)					
	Upper Fermented Layers of Grain		Middle Fermented Layers of Grain		Lower Fermented Layers of Grain	
	MSP-U	SSP-U	MSP-M	SSP-M	MSP-L	SSP-L
Ethyl methanoate	235.08 ± 94.66 ^b^	459.76 ± 160.27 ^a^	210.18 ± 114.29 ^a^	402.28 ± 157.81 ^a^	207.08 ± 122.88 ^a^	224.87 ± 108.87 ^a^
Ethyl acetate	1812.1 ± 223.84 ^b^	2460.59 ± 223.35 ^a^	2682.33 ± 1107.97^b^	2840.74 ± 295.97 ^a^	2774.64 ± 1005.81 ^a^	2795.56 ± 585.39 ^a^
Ethyl butyrate	487.92 ± 262.35 ^a^	593.7 ± 154.07 ^a^	530.99 ± 440.44 ^a^	614.29 ± 228.29 ^a^	620.86 ± 531.88 ^a^	685.86 ± 251.58 ^a^
1-Propanol	487.67 ± 255.75 ^a^	695.46 ± 330.64 ^a^	467.59 ± 284.17 ^a^	1232.77 ± 559.95 ^a^	612.69 ± 300.90 ^a^	1482.69 ± 720.01 ^a^
Isobutanol	184.2 ± 123.96 ^a^	175.86 ± 94.34 ^a^	166.15 ± 121.18 ^a^	206.52 ± 94.86 ^a^	179.86 ± 111.97 ^a^	185.29 ± 72.20 ^a^
Isoamyl acetate	15.9 ± 13.64 ^a^	16.73 ± 15.12 ^a^	19.91 ± 14.79 ^a^	23.79 ± 14.68 ^a^	20.51 ± 13.93 ^a^	19.25 ± 17.82 ^a^
Ethyl valerate	179.43 ± 89.42 ^a^	235.58 ± 80.32 ^a^	215.82 ± 136.95 ^a^	297.52 ± 114.66 ^a^	333.7 ± 280.75 ^b^	408.28 ± 143.84 ^a^
1-Butanol	496.08 ± 365.32 ^a^	387.54 ± 250.84 ^a^	443.94 ± 252.79 ^a^	647.92 ± 193.42 ^a^	528.92 ± 194.01 ^a^	970.99 ± 429.79 ^a^
3-Methyl-1-butanol	457.33 ± 230.13 ^a^	301.37 ± 20.68 ^a^	418.17 ± 179.38 ^a^	457.01 ± 109.57 ^a^	459.55 ± 178.81 ^a^	451.23 ± 236.07 ^b^
Ethyl caproate	2806.73 ± 790.85 ^a^	2091.37 ± 678.98 ^a^	2311.41 ± 826.22 ^a^	2230.49 ± 537.40 ^a^	2250.15 ± 1245.53 ^a^	3350.5 ± 706.54 ^a^
1-Pentanol	26.92 ± 22.77 ^a^	14.7 ± 11.53 ^a^	24.78 ± 15.11 ^b^	28.79 ± 10.15 ^a^	30.3 ± 12.57 ^a^	57.9 ± 23.43 ^a^
Caproic acid propyl ester	4.49 ± 3.25 ^a^	2.72 ± 2.07 ^a^	4.37 ± 3.27 ^a^	6.45 ± 3.64 ^a^	6.88 ± 5.68 ^a^	11.66 ± 4.72 ^a^
Ethyl heptanoate	133.01 ± 70.94 ^a^	104.32 ± 37.16 ^a^	106.9 ± 35.15 ^a^	129.11 ± 49.92 ^a^	166.33 ± 127.38 ^a^	250.86 ± 63.04 ^a^
Ethyl L(-)-lactate	698.37 ± 166.02 ^b^	889.84 ± 262.05 ^a^	821.9 ± 270.35 ^a^	1041.07 ± 240.47 ^a^	800.14 ± 262.51 ^a^	1159.62 ± 572.36 ^a^
1-Hexanol	115.9 ± 92.00 ^a^	48.07 ± 27.13 ^a^	102.79 ± 66.56 ^a^	109.19 ± 27.98 ^a^	117.42 ± 57.81 ^a^	222.91 ± 32.37 ^a^
Hexanoic acid butyl ester	46.65 ± 43.72 ^a^	13.05 ± 7.96 ^a^	33.22 ± 19.95 ^a^	34.55 ± 18.97 ^a^	55.29 ± 60.12 ^a^	72.08 ± 27.32 ^a^
Hexyl butyrate	20.86 ± 51.43 ^a^	2.26 ± 2.13 ^a^	13 ± 30.19 ^a^	6.12 ± 4.24 ^a^	10.52 ± 27.23 ^a^	9.38 ± 4.15 ^a^
Ethyl caprylate	70.19 ± 50.17 ^a^	46.78 ± 24.12 ^a^	49.82 ± 16.21 ^b^	65.58 ± 26.25 ^a^	56.82 ± 19.46 ^a^	101.9 ± 30.50 ^a^
Acetic acid glacial	613.8 ± 253.58 ^a^	393.59 ± 133.25 ^a^	569.14 ± 260.46 ^a^	514.43 ± 140.11 ^a^	574.06 ± 209.52 ^a^	723.9 ± 333.80 ^a^
Propionic acid	3.61 ± 6.46 ^a^	2.01 ± 5.01 ^a^	4.57 ± 5.43 ^a^	7.83 ± 11.86 ^a^	6.79 ± 6.36 ^a^	11.28 ± 16.79 ^a^
Butyric acid	197.26 ± 137.90 ^a^	94.28 ± 48.82 ^a^	238.03 ± 229.72 ^a^	166.7 ± 144.17 ^a^	227.31 ± 180.59 ^a^	538.13 ± 315.81 ^a^
Ethyl phenylacetate	2.66 ± 7.69 ^a^	0.25 ± 0.32 ^a^	5.09 ± 16.06 ^a^	0.22 ± 0.59 ^a^	1.63 ± 3.07 ^a^	1.1 ± 1.11 ^a^
Hexanoic acid	121.47 ± 76.72 ^a^	69.2 ± 54.61 ^a^	115.84 ± 86.22 ^a^	116.15 ± 115.78 ^a^	118.64 ± 76.84 ^a^	508.84 ± 298.93 ^a^
Heptanoic acid	3.77 ± 7.54 ^a^	1.43 ± 2.70 ^a^	1.79 ± 2.30 ^a^	2.63 ± 4.66 ^a^	2.09 ± 2.20 ^a^	10.35 ± 11.70 ^a^
Ethyl myristate	62.89 ± 127.36 ^a^	8.72 ± 7.65 ^a^	114.87 ± 301.70 ^a^	12.06 ± 9.52 ^a^	64.62 ± 130.19 ^a^	7.32 ± 6.27 ^a^
Octanoic acid	14.36 ± 24.04 ^a^	6.73 ± 11.15 ^a^	4.9 ± 6.65 ^b^	10.62 ± 15.82 ^a^	5.89 ± 5.70 ^a^	14.63 ± 15.21 ^a^
Ethyl hexadecanoate	94.04 ± 135.03 ^a^	100.18 ± 103.25 ^a^	72.32 ± 117.70 ^a^	291.71 ± 324.12 ^a^	74.6 ± 109.84 ^a^	122.58 ± 178.81 ^a^
Ethyl linoleate	282.67 ± 219.41 ^a^	322.93 ± 238.47 ^a^	255.41 ± 222.77 ^a^	637.44 ± 467.96 ^a^	359.73 ± 324.58 ^a^	312.84 ± 238.30 ^a^

Values are means ± SD. Within the same fermented layers of grain, the different letters ^(a,b)^ in the same row indicate the values are significantly different (*p* ≤ 0.05) (MSP-U and SSP-U; MSP-M and SSP-M; MSP-L and SSP-L). MSP: mud-sealing pits, SSP: steel-sealing pits; U: upper layer of fermentation grains, M: middle layer of fermentation grains, L: lower layer of fermentation grains.

**Table 3 foods-12-02579-t003:** The concentration of flavor compounds in the upper, middle, and lower layers of the second segment of strongly flavored Baijiu in the pit under two pit sealing methods.

Flavor Substances	Flavor Substance Concentration (mg·L^−1^)					
	Upper Fermented Layers of Grain		Middle Fermented Layers of Grain		Lower Fermented Layers of Grain	
	MSP-U	SSP-U	MSP-M	SSP-M	MSP-L	SSP-L
Ethyl methanoate	71.03 ± 36.90 ^a^	153.65 ± 78.64 ^a^	70.44 ± 33.68 ^a^	114.28 ± 26.39 ^a^	71.23 ± 29.80 ^a^	123.05 ± 35.46 ^a^
Ethyl acetate	999.44 ± 375.79 ^b^	2330.36 ± 399.78 ^a^	1410.46 ± 268.49 ^a^	1711.45 ± 163.05 ^a^	1625.14 ± 391.32 ^a^	2082.55 ± 287.88 ^a^
Ethyl butyrate	153.53 ± 108.87 ^a^	241.96 ± 105.75 ^a^	151.85 ± 68.44 ^a^	243 ± 87.13 ^a^	196.89 ± 93.52 ^a^	402.86 ± 112.21 ^a^
1-Propanol	341.22 ± 193.66 ^a^	659.21 ± 369.21 ^a^	340.63 ± 191.69 ^b^	1016.33 ± 481.07 ^a^	469.38 ± 177.48 ^a^	1202.15 ± 432.57 ^a^
Isobutanol	96.5 ± 75.83 ^a^	97.4 ± 51.01 ^a^	80.28 ± 55.70 ^a^	111.07 ± 51.01 ^a^	99.41 ± 61.61 ^a^	127.57 ± 50.69 ^a^
Isoamyl acetate	8.21 ± 6.81 ^a^	10.46 ± 7.14 ^a^	7.79 ± 6.11 ^a^	6.71 ± 4.37 ^a^	11.69 ± 12.40 ^a^	7.62 ± 5.19 ^a^
Ethyl valerate	74.92 ± 57.86 ^b^	139.75 ± 60.19 ^a^	88.31 ± 47.28 ^a^	144.5 ± 63.79 ^a^	134.61 ± 64.47 ^a^	286.59 ± 92.00 ^a^
1-Butanol	400.72 ± 320.88 ^a^	382.45 ± 206.92 ^a^	387.6 ± 286.48 ^a^	579.2 ± 155.23 ^a^	483.57 ± 247.42 ^a^	855.96 ± 238.19 ^a^
3-Methyl-1-butanol	362.9 ± 205.83 ^a^	312.22 ± 157.90 ^a^	340.95 ± 192.15 ^a^	376.7 ± 93.57 ^a^	368.99 ± 181.08 ^a^	413.29 ± 136.30 ^a^
Ethyl caproate	1666.63 ± 551.24 ^a^	1345.67 ± 662.87 ^a^	1875.86 ± 681.75 ^a^	1269.1 ± 319.82 ^a^	2231.09 ± 1083.01 ^a^	2651.09 ± 649.99 ^a^
1-Pentanol	21.35 ± 16.66 ^a^	14.92 ± 10.17 ^a^	21.18 ± 14.72 ^a^	29.27 ± 10.18 ^a^	28.99 ± 13.60 ^a^	51.33 ± 9.22 ^a^
Caproic acid propyl ester	1.51 ± 1.73 ^a^	1.39 ± 1.12 ^a^	1.61 ± 1.35 ^a^	3.73 ± 3.22 ^a^	3.44 ± 2.56 ^a^	7.71 ± 2.37 ^a^
Ethyl heptanoate	57.95 ± 47.71 ^a^	56.22 ± 24.48 ^a^	51.82 ± 21.13 ^a^	65.99 ± 31.50 ^a^	92.61 ± 64.03 ^a^	183.83 ± 72.35 ^a^
Ethyl L(-)-lactate	1359.5 ± 452.66^b^	1462.16 ± 490.25 ^a^	1495.8 ± 410.03 ^a^	1844.52 ± 338.16 ^a^	1504.32 ± 489.43 ^b^	2008.2 ± 256.74 ^a^
1-Hexanol	117.51 ± 103.84 ^a^	73.23 ± 43.14 ^a^	117.26 ± 93.38 ^a^	144.95 ± 52.89 ^a^	139.59 ± 94.40 ^a^	244.64 ± 39.02 ^a^
Hexanoic acid butyl ester	15.19 ± 14.05 ^a^	6.4 ± 3.21 ^a^	14.02 ± 8.89 ^a^	16.62 ± 11.87 ^a^	30.89 ± 31.05 ^a^	57.74 ± 21.92 ^a^
Hexyl butyrate	7.79 ± 15.21 ^a^	0.57 ± 0.35 ^a^	5.93 ± 2.31 ^a^	2.95 ± 2.61 ^a^	4.76 ± 10.67 ^a^	3.4 ± 3.29 ^a^
Ethyl caprylate	27.34 ± 14.92 ^a^	24.26 ± 10.92 ^a^	27.16 ± 16.75 ^a^	31.31 ± 18.78 ^a^	43.78 ± 30.29 ^a^	74.38 ± 28.54 ^a^
Acetic acid glacial	554.36 ± 165.33 ^a^	459.66 ± 149.95 ^a^	572.94 ± 153.13 ^b^	650.8 ± 132.11 ^a^	573.73 ± 114.71 ^a^	789.84 ± 206.79 ^a^
Propionic acid	15.04 ± 16.58 ^a^	4.65 ± 3.11 ^a^	19.63 ± 18.06 ^a^	12.57 ± 15.23 ^a^	22.35 ± 18.44 ^a^	10.16 ± 10.78 ^a^
Butyric acid	171.91 ± 127.64 ^a^	70.78 ± 35.55 ^a^	210.31 ± 166.83 ^a^	246.54 ± 214.47 ^a^	335.86 ± 252.56 ^a^	606.57 ± 211.58 ^a^
Ethyl phenylacetate	0.86 ± 1.34 ^a^	1.06 ± 2.01 ^a^	1.09 ± 1.65 ^a^	0.67 ± 1.19 ^a^	1.3 ± 1.70 ^a^	0.88 ± 1.30 ^a^
Hexanoic acid	131.32 ± 137.83 ^a^	27.86 ± 36.37 ^a^	118.3 ± 86.30 ^a^	166.11 ± 142.20 ^a^	225.5 ± 213.02 ^a^	546.31 ± 160.21 ^a^
Heptanoic acid	1.94 ± 3.19 ^a^	0.15 ± 0.52 ^a^	1.4 ± 1.21 ^a^	1.91 ± 3.20 ^a^	4.11 ± 2.65 ^a^	8.91 ± 7.78 ^a^
Ethyl myristate	5.14 ± 3.21 ^a^	1.8 ± 1.18 ^a^	6.12 ± 11.60 ^a^	1.38 ± 1.02 ^a^	5.03 ± 3.25 ^a^	1.05 ± 1.16 ^a^
Octanoic acid	4.4 ± 63.22 ^a^	0.74 ± 1.28 ^a^	2.92 ± 3.27 ^a^	2.79 ± 3.36 ^a^	5.04 ± 2.22 ^a^	10.39 ± 9.08 ^a^
Ethyl hexadecanoate	15.65 ± 22.58 ^a^	25.64 ± 25.66 ^a^	14.19 ± 18.79 ^a^	23.22 ± 22.10 ^a^	16.18 ± 22.82 ^b^	17.29 ± 18.80 ^a^
Ethyl linoleate	37.32 ± 24.67 ^a^	41.37 ± 23.08 ^a^	31.42 ± 21.40 ^a^	45.15 ± 16.51 ^a^	35.46 ± 23.85 ^a^	40.24 ± 16.88 ^a^

Values are means ± SD. Within the same fermented layers of grain, the different letters ^(a,b)^ in the same row indicate the values are significantly different (*p* ≤ 0.05) (MSP-U and SSP-U; MSP-M and SSP-M; MSP-L and SSP-L). MSP: mud-sealing pits, SSP: steel-sealing pits; U: upper layer of fermentation grains, M: middle layer of fermentation grains, L: lower layer of fermentation grains.

**Table 4 foods-12-02579-t004:** The concentration of flavor compounds in the upper, middle, and lower layers of the third segment of strongly flavored Baijiu in the pit under two pit sealing methods.

Flavor Substances	Flavor Substance Concentration (mg·L^−1^)					
	Upper Fermented Layers of Grain		Middle Fermented Layers of Grain		Lower Fermented Layers of Grain	
	MSP-U	SSP-U	MSP-M	SSP-M	MSP-L	SSP-L
Ethyl methanoate	16.09 ± 16.37 ^a^	15.73 ± 12.02 ^a^	16.75 ± 12.32 ^a^	21.62 ± 19.52 ^a^	19.31 ± 16.24 ^a^	23.79 ± 21.03 ^a^
Ethyl acetate	226.64 ± 99.49 ^b^	421.03 ± 139.23 ^a^	522 ± 193.52 ^a^	481.92 ± 96.07 ^a^	645.09 ± 202.73 ^a^	672.14 ± 131.17 ^a^
Ethyl butyrate	48.52 ± 28.48 ^a^	42.49 ± 27.37 ^a^	55 ± 26.82 ^a^	77.64 ± 41.53 ^a^	75.72 ± 37.57 ^a^	156.69 ± 64.26 ^a^
1-Propanol	213.58 ± 113.31 ^a^	404.09 ± 193.22 ^a^	193.76 ± 115.14 ^a^	635.73 ± 320.90 ^a^	285.78 ± 117.20 ^a^	877.9 ± 404.97 ^a^
Isobutanol	39.38 ± 35.98 ^a^	35.55 ± 22.26 ^a^	26.29 ± 9.03 ^a^	40.5 ± 22.06 ^a^	43.27 ± 34.75 ^a^	59.81 ± 30.84 ^a^
Isoamyl acetate	2.65 ± 2.81 ^a^	2.77 ± 3.22 ^a^	2.62 ± 1.32 ^b^	3.94 ± 3.00 ^a^	3.04 ± 2.65 ^a^	4.38 ± 3.44 ^a^
Ethyl valerate	32.36 ± 24.54 ^a^	38.41 ± 24.32 ^a^	44.17 ± 26.56 ^a^	55.62 ± 27.94 ^a^	60.19 ± 30.55 ^b^	122.29 ± 47.98 ^a^
1-Butanol	249.67 ± 209.21 ^a^	250.4 ± 130.12 ^a^	238.58 ± 190.40 ^a^	356.75 ± 113.81 ^a^	306.03 ± 188.72 ^a^	541.3 ± 184.70 ^a^
3-Methyl-1-butanol	199.22 ± 125.42 ^a^	195.4 ± 57.62 ^a^	190.19 ± 128.97 ^a^	208.92 ± 63.11 ^a^	214.53 ± 125.64 ^a^	247.12 ± 73.97 ^a^
Ethyl caproate	905.53 ± 346.18 ^a^	484.1 ± 149.54 ^a^	1012.77 ± 281.10 ^a^	648.4 ± 220.26 ^a^	1265.07 ± 541.02 ^a^	1570.82 ± 365.85 ^a^
1-Pentanol	13.74 ± 10.48 ^a^	13.18 ± 5.02 ^a^	13.79 ± 10.08 ^b^	19.28 ± 6.20 ^a^	19.5 ± 10.88 ^a^	37.15 ± 7.31 ^a^
Caproic acid propyl ester	0.73 ± 1.08 ^a^	0.53 ± 0.24 ^a^	0.74 ± 0.54 ^a^	1.49 ± 1.33 ^a^	1.66 ± 1.33 ^a^	4.17 ± 1.75 ^a^
Ethyl heptanoate	32.51 ± 27.14 ^a^	26.38 ± 11.54 ^a^	30.95 ± 13.82 ^a^	34.45 ± 12.33 ^a^	59.17 ± 45.12 ^a^	115.49 ± 47.20 ^a^
Ethyl L(-)-lactate	2648.81 ± 1229.76 ^a^	2940.37 ± 628.09 ^a^	2842.17 ± 933.37 ^a^	2920.39 ± 552.66 ^a^	2521.66 ± 864.35 ^a^	2616.47 ± 383.57 ^a^
1-Hexanol	92.78 ± 81.72 ^a^	91.24 ± 33.48 ^b^	93.33 ± 77.25 ^b^	116.73 ± 31.10 ^a^	117.48 ± 92.38 ^a^	192.08 ± 65.62 ^a^
Hexanoic acid butyl ester	13.12 ± 13.55 ^a^	4.32 ± 2.46 ^a^	11.95 ± 7.32 ^a^	11.38 ± 5.00 ^a^	22.94 ± 20.95 ^a^	41.43 ± 12.75 ^a^
Hexyl butyrate	23.36 ± 16.25 ^a^	0 ± 0.00 ^a^	19.11 ± 8.65 ^a^	0.43 ± 1.15 ^a^	17.32 ± 10.25 ^a^	1.82 ± 1.58 ^a^
Ethyl caprylate	29.04 ± 26.32 ^a^	17.01 ± 6.20 ^a^	27.73 ± 21.94 ^a^	20.69 ± 8.05 ^a^	56.31 ± 36.21 ^a^	55.46 ± 20.42 ^a^
Acetic acid glacial	734.36 ± 241.57 ^a^	693.69 ± 142.03 ^a^	808.87 ± 257.65 ^a^	899.4 ± 228.82 ^a^	745.66 ± 218.01 ^a^	958.74 ± 222.63 ^a^
Propionic acid	24.52 ± 12.32 ^a^	11.03 ± 25.44 ^a^	24.99 ± 20.21 ^a^	17.77 ± 35.31 ^a^	26.54 ± 21.02 ^a^	24.74 ± 19.67 ^a^
Butyric acid	347.55 ± 272.00 ^a^	172.21 ± 59.24 ^a^	454.97 ± 437.84 ^a^	401.59 ± 319.99 ^a^	625.4 ± 236.27 ^a^	848.57 ± 237.99 ^a^
Ethyl phenylacetate	2.3 ± 2.95 ^a^	1.51 ± 1.20 ^a^	2.51 ± 1.35 ^a^	1.29 ± 1.02 ^a^	2.31 ± 2.01 ^a^	1.67 ± 1.06 ^a^
Hexanoic acid	235.35 ± 210.98 ^a^	112.99 ± 41.17 ^a^	253.99 ± 175.17 ^a^	259.1 ± 177.62 ^a^	345.06 ± 269.05 ^a^	779.06 ± 172.83 ^a^
Heptanoic acid	4.85 ± 4.25 ^a^	1.35 ± 1.20 ^a^	3.84 ± 3.59 ^a^	2.89 ± 2.70 ^a^	7.51 ± 11.84 ^a^	13.04 ± 9.55 ^a^
Ethyl myristate	6.88 ± 2.52 ^a^	2.32 ± 2.01 ^a^	8.38 ± 2.68 ^a^	2.32 ± 1.12 ^a^	6.95 ± 2.68 ^a^	1.74 ± 0.89 ^a^
Octanoic acid	9.12 ± 5.21 ^a^	1.63 ± 1.00 ^a^	7.02 ± 7.28 ^a^	3.59 ± 2.21 ^a^	9.91 ± 8.47 ^a^	15.97 ± 12.25 ^a^
Ethyl hexadecanoate	22.01 ± 2.23 ^b^	30.1 ± 25.61 ^a^	28.14 ± 33.73 ^a^	29.49 ± 25.03 ^a^	25.69 ± 19.66 ^b^	33.51 ± 30.27 ^a^
Ethyl linoleate	37.18 ± 21.02 ^a^	42.76 ± 6.72 ^a^	42.7 ± 30.88 ^a^	44.43 ± 7.46 ^a^	40.53 ± 28.81 ^a^	47.06 ± 17.54 ^a^

Values are means ± SD. Within the same fermented layers of grain, the different letters ^(a,b)^ in the same row indicate the values are significantly different (*p* ≤ 0.05) (MSP-U and SSP-U; MSP-M and SSP-M; MSP-L and SSP-L). MSP: mud-sealing pits, SSP: steel-sealing pits; U: upper layer of fermentation grains, M: middle layer of fermentation grains, L: lower layer of fermentation grains.

The comparison of the four major esters and three major acids of Baijiu in the 1st, 2nd, and 3rd segment of the upper, middle, and lower layers of fermented grains (Table 5 and Table 6) is as follows:

For the distilling segment of the upper layer of fermented grains, the concentration of each of hexanoic acid, butyric acid, and acetic acid in the traditional mud pit sealing method was higher than that under the steel pit sealing method. Except for ethyl hexanoate, the total concentration of each of ethyl lactate, ethyl butyrate, and ethyl acetate was lower than that of the strongly flavored Baijiu under the steel pit sealing method. The possible reason for these results was that the upper layer of fermented grains had greater contact with the sealing mud under the mud sealing method, which resulted in a higher concentration of acid substances required for the later synthesis of esters compared with the steel sealing method.

For the middle layer of fermented grains, the concentration of caproic acid and acetic acid under the steel pit sealing method was higher than that under the traditional mud pit sealing method, and the total concentration of ethyl lactate, ethyl caproate, ethyl butyrate, and ethyl acetate was higher than that under the traditional pit sealing method. The fermentation of the middle layer of fermented grains in the pit was better than that of the upper layer of fermented grains, which may have been due to the better sealing performance under the steel pit sealing method.

For the distilling segment of the lower fermented grains, the concentration of acetic acid, butyric acid, caproic acid, ethyl acetate, ethyl caproate, ethyl lactate, and ethyl butyrate under the steel pit sealing method was each higher than those under the traditional mud pit sealing method. Over time, the advantages of the steel pit sealing method became apparent because it had better sealing performance inside the pit and was less prone to pollution, which resulted in a relatively higher concentration of esters.

### 3.4. Differentiation of Strongly Flavored Baijiu with Two Different Pit Sealing Methods by HCA

HCA is a technique for natural aggregation of research objects by similarity, an unsupervised pattern recognition method, and a statistical analysis technique for classifying research objects into relatively homogeneous groups [37,38]. In this experiment, 28 volatile compounds with an area percentage > 0.5% were selected, and their concentrations were used as a variable to standardize the data. The strongly flavored Baijiu with the 1st, 2nd, and 3rd segment of the two sealing methods in the upper, middle, and lower strong flavor Baijiu were clustered into two categories, with the mud-sealing Baijiu samples divided into one category and the steel-sealing Baijiu samples clustered into another category (Figure 7). The differences between the sealing methods indicated that the clustering analysis differentiated the strongly flavored Baijiu produced by the different sealing methods and that the 28 quantitative substances selected were correct. However, the HCA method did not fully cluster the same batch of Baijiu together. For example, MGT 141, MGT 241, and MGT 341 were divided into two clusters in the top lees of the Baijiu sample. A possible reason for this is that the mud-sealing pit was not stable due to external influences during the brewing process.

### 3.5. Analysis of PCA Scores for Two Types of Pit Sealing Methods

PCA is a multivariate statistical analysis method in which multiple variables are transformed linearly in an unsupervised mode to select a smaller number of significant variables [39,40]. Using SPSS 23.0, PCA was performed on 28 volatile compounds of strong flavor Baijiu to extract data with eigenvalues > 1, and the original information was reduced to seven principal components. The variance contributions of F1, F2, F3, F4, F5, F6, and F7 were 30.433%, 15.441%, 11.520%, 5.452%, 4.984%, 4.413%, and 3.737%, respectively, with a cumulative variance contribution of 75.979% (Table 7). This indicated that these seven principal components explained 75.979% of the 28 variables. Taking the first seven principal components for analysis was feasible due to the large sample size and the cumulative contribution that was close to 80%, which ensured that the condensed composite variables were representative of most of the data. The 28 original variables of strong flavor base Baijiu with different pit sealing methods (ethyl valerate, ethyl heptate, propyl caproate… ethyl lactate, and propionic acid are denoted as X_1_, X_2_, X_3_ … X_27_, X_28_, respectively) had a linear relationship with each principal component. Table 8 shows the coefficients of the relationships between each variable and the principal components, and the linear equations for the seven principal components and variables are as follows:
F1 = 0.299X_1_ + 0.293X_2_ + 0.287X_3_ + 0.282X_4_ + 0.268X_5_ + 0.237X_6_ + 0.234X_7_ + 0.234X_8_ + 0.233X_9_ + 0.226X_10_ + 0.215X_11_ + 0.212X_12_ + 0.202X_13_ + 0.202X_14_ + 0.190X_15_ + 0.086X_16_ − 0.006X_17_ + 0.123X_18_ − 0.052X_19_ − 0.044X_20_ + 0.087X_21_ − 0.022X_22_ + 0.137X_23_ + 0.119X_24_ + 0.161X_25_ + 0.068X_26_ − 0.090X_27_ − 0.060X_28_F2 = 0.009X_1_ + 0.092X_2_ + 0.147X_3_ − 0.077X_4_ − 0.194X_5_ − 0.220X_6_ + 0.167X_7_ − 0.022X_8_ + 0.120X_9_ + 0.206X_10_ + 0.049X_11_ − 0.197X_12_ + 0.083X_13_ − 0.195X_14_ − 0.172X_15_ + 0.399X_16_ + 0.314X_17_ + 0.293X_18_ + 0.283X_19_ + 0.096X_20_ + 0.023X_21_ + 0.028X_22_ + 0.261X_23_ − 0.095X_24_ − 0.169X_25_ + 0.227X_26_ + 0.236X_27_ + 0.175X_28_F3 = 0.145X_1_ + 0.097X_2_ + 0.051X_3_ + 0.143X_4_ + 0.070X_5_ + 0.096X_6_ + 0.213X_7_ + 0.132X_8_ − 0.238X_9_ − 0.203X_10_ − 0.108X_11_ − 0.050X_12_ − 0.007X_13_ + 0.041X_14_ − 0.261X_15_ − 0.063X_16_ + 0.140X_17_ + 0.051X_18_ − 0.021X_19_ + 0.445X_20_ + 0.430X_21_ + 0.382X_22_ − 0.309X_23_ − 0.142X_24_ − 0.066X_25_ − 0.100X_26_ + 0.038X_27_ + 0.017X_28_F4 = −0.138X_1_ − 0.007X_2_ − 0.040X_3_ − 0.174X_4_ − 0.070X_5_ + 0.008X_6_ + 0.037X_7_ + 0.133X_8_ + 0.018X_9_ − 0.029X_10_ + 0.018X11 − 0.345X_12_ + 0.096X_13_ − 0.126X_14_ − 0.079X_15_ − 0.059X_16_ + 0.121X_17_ − 0.030X_18_ − 0.227X_19_ + 0.088X_20_ + 0.030X_21_ + 0.202X_22_ + 0.040X_23_ + 0.584X_24_ + 0.431X_25_ + 0.284X_26_ − 0.200X_27_ + 0.073X_28_F5 = −0.005X_1_ + 0.030X_2_ − 0.005X_3_ − 0.132X_4_ − 0.074X_5_ − 0.113X_6_ + 0.127X_7_ − 0.169X_8_ + 0.321X_9_ + 0.253X_10_ + 0.060X_11_ − 0.085X_12_ + 0.131X_13_ − 0.090X_14_ + 0.188X_15_ − 0.229X_16_ − 0.001X_17_ − 0.267X_18_ − 0.313X_19_ + 0.154X_20_ + 0.103X_21_ + 0.293X_22_ + 0.252X_23_ + 0.054X_24_ − 0.226X_25_ − 0.42X_26_ + 0.203X_27_ + 0.007X_28_F6 = 0.065X_1_ + 0.006X_2_ + 0.002X_3_ + 0.012X_4_ + 0.066X_5_ + 0.019X_6_ − 0.058X_7_ − 0.275X_8_ − 0.060X_9_ − 0.072X_10_ + 0.078X_11_ + 0.001X_12_ − 0.154X_13_ + 0.273X_14_ + 0.087X_15_ − 0.009X_16_ − 0.174X_17_ + 0.186X_18_ + 0.120X_19_ + 0.030X_20_ + 0.077X_21_ − 0.040X_22_ − 0.063X_23_ + 0.280X_24_ + 0.266X_25_ − 0.160X_26_ + 0.514X_27_ + 0.511X_28_F7 = 0.001X_1_ − 0.141X_2_ − 0.014X_3_ + 0.157X_4_ − 0.005X_5_ + 0.098X_6_ − 0.229X_7_ + 0.084X_8_ + 0.040X_9_ + 0.067X_10_ + 0.297X_11_ + 0.181X_12_ + 0.168X_13_ − 0.032X_14_ − 0.066X_15_ − 0.066X_16_ + 0.368X_17_ − 0.354X_18_ + 0.376X_19_ + 0.082X_20_ − 0.383X_21_ + 0.312X_22_ − 0.145X_23_ − 0.011X_24_ + 0.154X_25_ − 0.071X_26_ + 0.094X_27_ + 0.095X_28_


The scoring model was F = 30.433F1 + 15.441F2 + 11.52F3 + 5.452F4 + 4.984F5 + 4.413F6 + 3.737F7, which we used to calculate the comprehensive score of strongly flavored Baijiu for different pit sealing methods (Table 9). In general, the average score of steel-sealing pits was higher than that of mud-sealing pits. The higher score of the strong flavor Baijiu with the steel cap pits may be attributed to the improvement in the sealing of the steel caps, which ensured the anaerobic environment in the pit to enhance the anaerobic respiration of microorganisms and metabolism of more flavor substances.

The PCA method can be applied to the discriminant function of product type and variety. We selected the samples of strongly flavored Baijiu produced from the upper, middle, and lower levels of fermented grains as the scatter diagram (Figure 8), and the results are as follows:

From the figure, the strongly flavored Baijiu produced from different pit sealing forms overlapped and could not be separated clearly. The reason may be that the sample size was too large and the fermentation process was not significantly different, which resulted in incomplete separation. Among them, the scatter diagram of the first segment Baijiu of the upper, middle, and lower levels of fermented grains was relatively scattered, which may be because the distillery usually relied on the experience of the master when picking Baijiu, and there was the possibility of picking Baijiu too early or too later. However, each layer of segmented Baijiu was obviously gathered together, which was consistent with the results obtained by HCA. This indicated that PCA and HCA were consistent with each other when classifying and identifying strongly flavored Baijiu with different pit sealing methods.

### 3.6. Discriminant Analysis

Five typical discriminant functions were obtained by discriminant analysis for each of the different layers of grains. The upper layer of grains with eigenvalues of 25.19, 11.00, 2.70, 1.37, and 0.44 explained 61.90%, 27.00%, 6.60%, 3.40%, and 1.10%, respectively, of the variation in the model. The first two typical discriminant analysis functions explained 88.90% of the variance. The middle layer of grains with eigenvalues of 18.03, 7.32, 5.04, 2.20, and 0.67 explained 54.20%, 22.00%, 15.10%, 6.60%, and 2.00%, respectively, of the variation in the model. The first two typical discriminant analysis functions explained 76.20% of the variation. The lower layer of grains with eigenvalues of 12.03, 5.32, 2.38, 1.91, and 0.29 explained 54.90%, 24.20%, 10.90%, 8.70%, and 1.30%, respectively, of the variation in the model. The first two typical discriminant analysis functions explained 79.10% of the variance.

It is possible to describe the differences and links between the flavor components in the top lees 1st, 2nd, and 3rd segments of strongly flavored Baijiu in mud-sealing pits and steel-sealing pits. The first two typical discriminant functions were used to make scatter plots for the different segment distillations of the same layer of fermented grains (Figure 9). For the corresponding 1st, 2nd, and 3rd segments Baijiu of the upper, middle, and lower layers of fermented grains, the centroids of the six types of Baijiu samples were separated from each other without any overlap. The group mass centers of the 1st, 2nd, and 3rd segments of the mud-sealing pits and steel-sealing pits were closer together, and the samples were more similar. However, the mud-sealing pit samples were separated approximately from the steel-sealing pit samples, but this result was in general agreement with the results of the principal component analysis, which indicated that both principal component analysis and discriminant analysis were very effective tools for the two different sealing methods. These statistical methods were useful for classification and prediction.

We analyzed 108 strongly flavored Baijiu samples from the upper, middle, and lower layers of fermented grains using DA. We randomly selected one sample from the 1st, 2nd, and 3rd segments of steel and mud sealing pit brewing as a “test sample”, which totaled six samples. We used the remaining 102 samples as “training samples” to establish a discriminant function and then discriminated the test samples to determine their classification to compare them with the actual classification of the samples. In this study, six typical variables were obtained, and a discriminant function was established based on these variables to classify two different types of strongly flavored Baijiu with different pit sealing methods, with an accurate discrimination rate of 100.0% (Table 10, Table 11 and Table 12).

Based on the predicted results of the discriminant analysis, we used the GC-MS analysis of the total ion flow chromatogram of strongly flavored Baijiu combined with the discriminant analysis to classify and to identify the model for the two different types of strongly flavored Baijiu with different pit sealing methods. The correct identification rate of the samples was 100.0%.

## 4. Conclusions

The types and the number of flavor substances produced by fermentation in mud-sealing and steel-sealing pit were relatively similar. In the PCA experiment, steel-sealing pits were more significant than traditional mud-sealing pits. HCA can divide samples into two categories, and using DA can also separate samples without aggregation. At the same time, two types of samples (one is mud-sealing and the other is steel-sealing) were successfully identified. At first, the relationship between the two methods is mutually verified. By this way, it can result in a more robust conclusion. Economically, the results of statistical analysis showed that the method of steel-sealing the pits increased the yield of Baijiu by 6.7% compared with the mud-sealing pit in the brewing process, with no significant difference in taste quality. Based on flavor and yield, both at the experimental and theoretical levels, we found no significant difference between the steel-sealing and the traditional mud-sealing final product (Baijiu) for Chinese Baijiu brewing. Compared with the disadvantages of the mud-sealing pits, where the yield was greatly influenced by the environment and was unstable, the process of steel sealing pits exhibited better controllability and stability, and it was more resistant to adverse conditions. In the future, the selection of pits cap sealing for intensive solid state fermentation process based on artificial intelligence and mechanization will present the trend of gradually replacing mud sealing with steel sealing.

## Figures and Tables

**Figure 1 foods-12-02579-f001:**
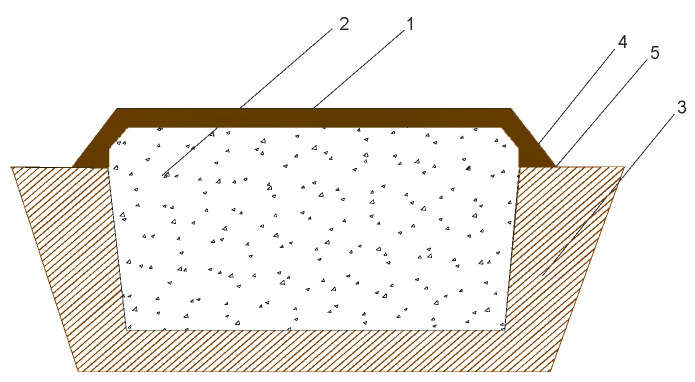
Profile of traditional fermentation device (pit cap: mud) for producing strongly flavored Baijiu in China. (1—mud-sealing cap; 2—Grains of Baijiu; 3—Brewing pit; 4—Sealing the pit opening; 5—Included angle of prism clamp, Modified from Figure 1 in Lu et al., 2012 [14]).

**Figure 2 foods-12-02579-f002:**
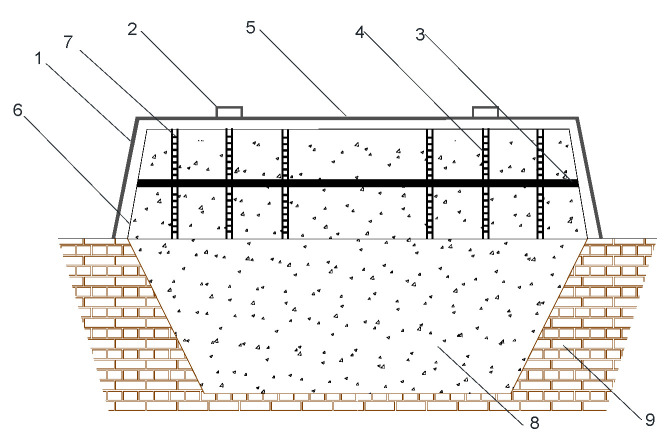
Profile of pit cap improved fermentation device A(pit cap: steel) for producing strongly flavored Baijiu in China. (1—steel-sealing cap; 2—Lifting ring; 3—Fixing brackets; 4—Pit mud board; 5—Cover top; 6—Lower box; 7—Pit mud; 8—Grains of Baijiu; 9—Brewing pit, Modified from Figure 2 in Zhang, 2015 [11]).

**Figure 3 foods-12-02579-f003:**
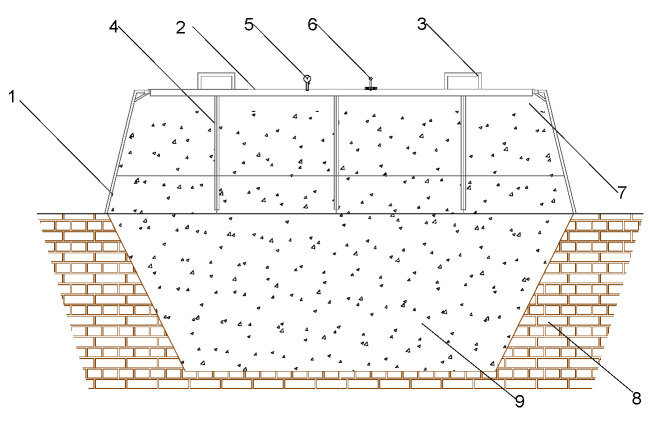
Profile of pit cap improved fermentation device B(pit cap: steel) for producing strongly flavored Baijiu in China. (1—Lower box; 2—Upper cover body; 3—Lifting ring; 4—Lower box side reinforcements; 5—Pressure Gauges; 6—Material handling ports; 7—Outer side wall of the lower box; 8-Brewing pit; 9—Grains of Baijiu, Modified from Figure 3 in Zhang and Zhao et al., 2015 [17]).

**Figure 4 foods-12-02579-f004:**
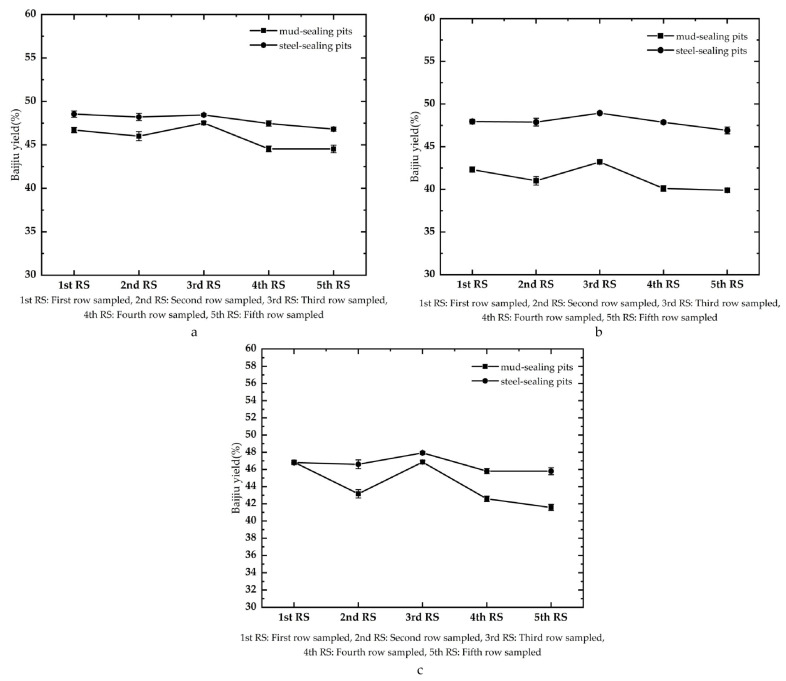
Variation in the yield of strongly flavored Baijiu from the same pit using mud sealing or steel sealing under five sampling batches. ((**a**)—Pit No. 1; (**b**)—Pit No. 2; (**c**)—Pit No. 3).

**Figure 5 foods-12-02579-f005:**
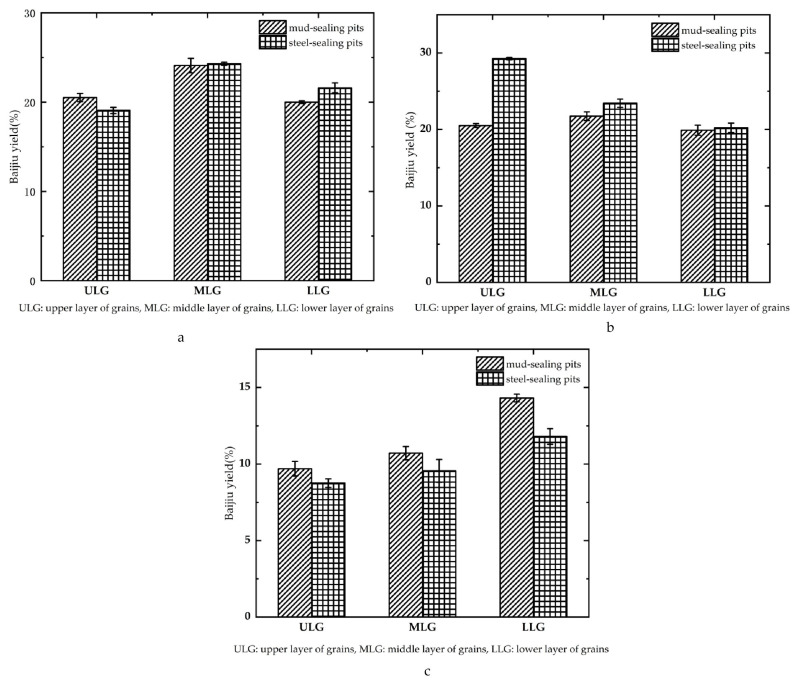
Yield of strongly flavored Baijiu from two different pit sealing methods. ((**a**)—1 segment of strong flavor Baijiu; (**b**)—2 segment of strong flavor Baijiu; (**c**)—3 segment of strong flavor Baijiu).

**Figure 6 foods-12-02579-f006:**
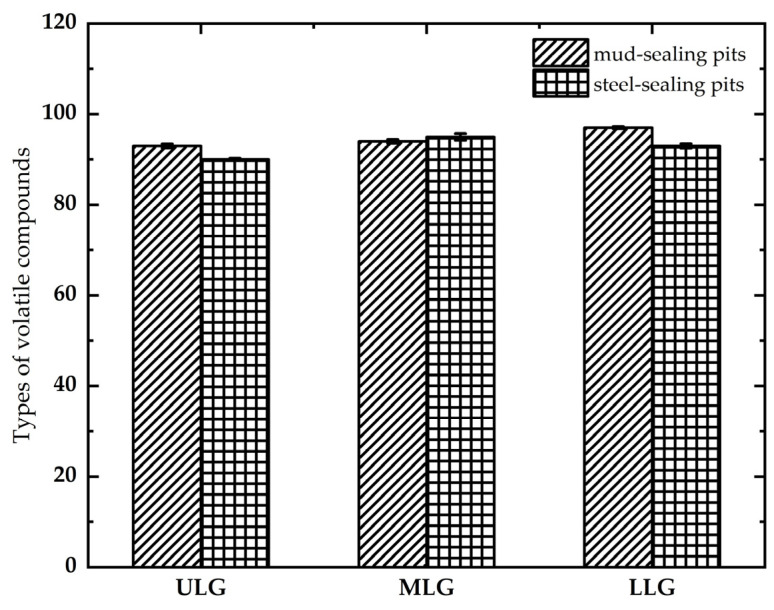
Amount of volatile flavor compounds in the different layers of fermented grains in the process of making strongly flavored Baijiu in China.

**Figure 7 foods-12-02579-f007:**
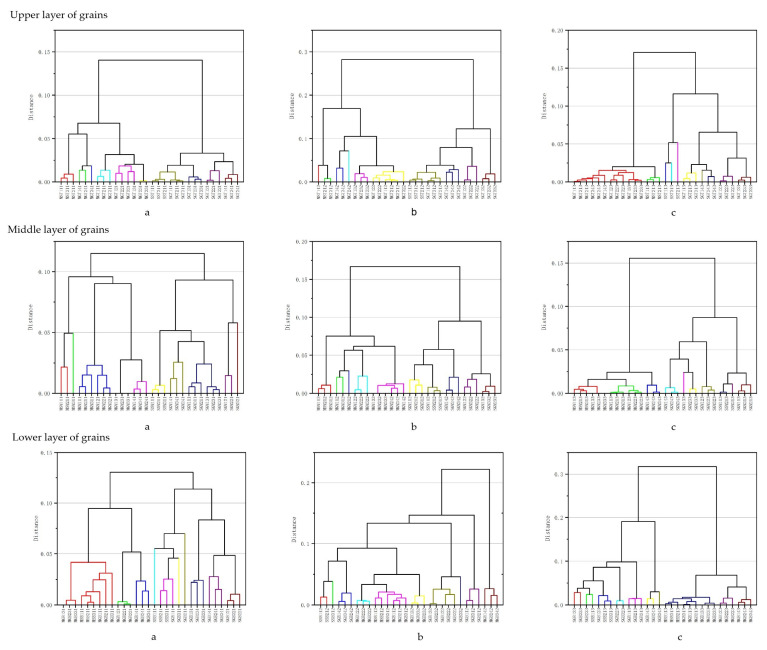
Clustering analysis of strongly flavored Baijiu with different pit sealing methods. ((**a**)—1 segment strong flavor Baijiu; (**b**)—2 segment strong flavor Baijiu; (**c**)—3 segment strong flavor Baijiu; The names of the letters in the above picture mean: initial “M” for mud-sealing pit, initial “S” for steel-sealing pit).

**Figure 8 foods-12-02579-f008:**
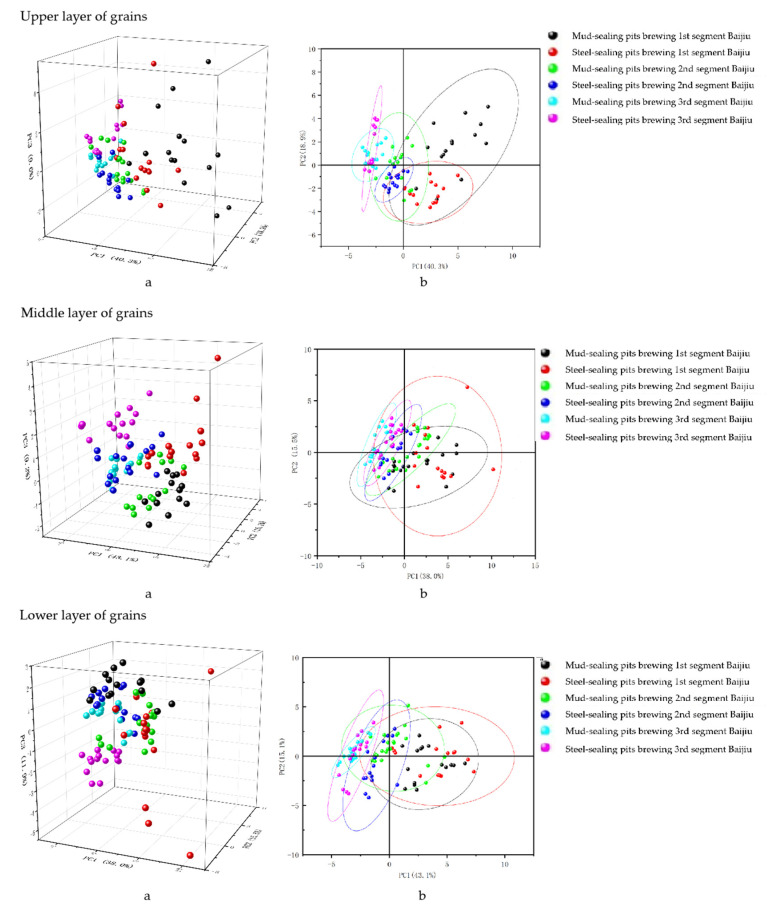
Scatter plot of the main components of the flavor substances of Baijiu. ((**a**)—three-dimensional scatter plot of PC1, PC2, and PC3; (**b**)—plan view of PC1 and PC2).

**Figure 9 foods-12-02579-f009:**
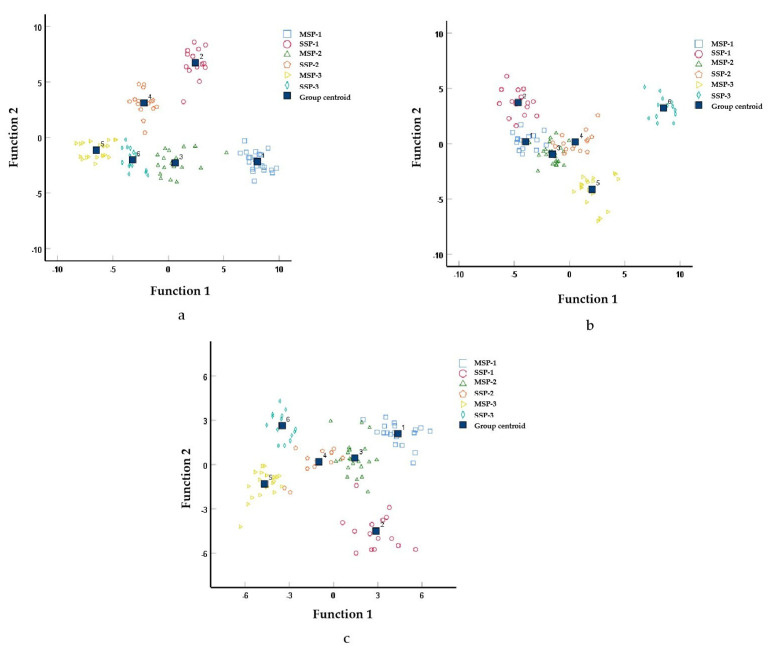
Scatter diagram of a typical discriminant analysis of the two sealing methods for strongly flavored Baijiu. ((**a**)—upper layer of grain; (**b**)—middle layer of grain; (**c**)—lower layer of grain; The names of the letters in the above picture mean: MSP for mud-sealing pits, SSP for steel-sealing pits; 1—1 segment strong flavor Baijiu; 2—2 segment strong flavor Baijiu; 3—3 segment strong flavor Baijiu).

**Table 5 foods-12-02579-t005:** Concentration of four major esters in different fermented layers of grain under two pit sealing methods.

Fermented Grains Layer	Pit Sealing Method	Ethyl Acetate (mg·L^−1^)	Ethyl Caproate (mg·L^−1^)	Ethyl Lactate (mg·L^−1^)	Ethyl Butyrate (mg·L^−1^)
Upper fermented grains	mud	1012.73 ± 5.86 ^b^	1792.96 ± 6.95 ^a^	1568.89 ± 9.31 ^a^	229.99 ± 14.32 ^b^
	steel	1737.33 ± 9.55 ^a^	1307.05 ± 8.20 ^a^	1764.12 ± 10.11 ^a^	292.72 ± 12.01 ^a^
Middle fermented grains	mud	1538.26 ± 10.21 ^a^	1733.35 ± 19.22 ^a^	1719.96 ± 15.20 ^a^	245.95 ± 8.95 ^a^
	steel	1678.04 ± 15.36 ^a^	1382.66 ± 14.32 ^b^	1935.33 ± 13.01 ^a^	311.64 ± 6.34 ^a^
Lower fermented grains	mud	1681.62 ± 7.21 ^a^	1915.44 ± 7.21 ^a^	1608.71 ± 16.62 ^a^	297.82 ± 7.41 ^a^
	steel	1850.08 ± 10.22 ^a^	2524.14 ± 6.74 ^a^	1928.10 ± 9.89 ^a^	415.14 ± 8.16 ^a^

Values are means ± SD. Within the same fermented layers of grain, the different letters ^(a,b)^ in the same column indicate the values are significantly different (*p* ≤ 0.05). (Comparing the upper/middle/lower layer of the mud- and steel-sealing pits, respectively).

**Table 6 foods-12-02579-t006:** Concentration of three major acids in different layers of fermented grain under two pit sealing methods.

Fermented Grains Layer	Pit Sealing Method	Acetic Acid (mg·L^−1^)	Butyrate (mg·L^−1^)	Caproic Acid (mg·L^−1^)
Upper fermented grains	mud	634.17 ± 7.25 ^a^	238.91 ± 3.21 ^a^	162.71 ± 7.20 ^a^
	steel	515.65 ± 6.68 ^a^	112.42 ± 10.35 ^a^	70.02 ± 9.51 ^b^
Middle fermented grains	mud	650.32 ± 8.69 ^a^	301.10 ± 12.27 ^a^	162.71 ± 4.32 ^a^
	steel	688.21 ± 10.21 ^a^	271.61 ± 6.63 ^a^	180.45 ± 5.88 ^a^
Lower fermented grains	mud	631.15 ± 15.64 ^a^	396.19 ± 7.86 ^a^	229.73 ± 11.02 ^a^
	steel	824.16 ± 9.44 ^a^	664.42 ± 3.28 ^a^	611.40 ± 12.11 ^a^

Values are means ± SD. Within the same fermented layers of grain, the different letters ^(a,b)^ in the same column indicate the values are significantly different (*p* ≤ 0.05). (Comparing the upper/middle/lower layer of the mud- and steel-sealing pits, respectively).

**Table 7 foods-12-02579-t007:** Eigenvalues and variance contribution of principal components for the analysis of the scores of two pit sealing methods.

Components	Eigenvalue	Percent Variance	Cumulative Contribution (%)
PC1	8.521	30.433	30.433
PC2	4.323	15.441	45.874
PC3	3.226	11.52	57.394
PC4	1.526	5.452	62.845
PC5	1.396	4.984	67.829
PC6	1.236	4.413	72.242
PC7	1.046	3.737	75.979

**Table 8 foods-12-02579-t008:** Composition matrix of 28 volatile compounds in strong flavor Baijiu based on principal component analysis.

Quantitative Compound	F1	F2	F3	F4	F5	F6	F7
Ethyl valerate (X_1_)	0.299	0.009	0.145	−0.138	−0.005	0.065	0.001
Ethyl heptanoate (X_2_)	0.293	0.092	0.097	−0.007	0.03	0.006	−0.141
Caproic acid propyl ester (X_3_)	0.287	0.147	0.051	−0.04	−0.005	0.002	−0.014
Ethyl butyrate (X_4_)	0.282	−0.077	0.143	−0.174	−0.132	0.012	0.157
Ethyl acetate (X_5_)	0.268	−0.194	0.07	−0.07	−0.074	0.066	−0.005
Ethyl methanoate (X_6_)	0.237	−0.22	0.096	0.008	−0.113	0.019	0.098
Hexanoic acid butyl ester (X_7_)	0.234	0.167	0.213	0.037	0.127	−0.058	−0.229
Ethyl caproate (X_8_)	0.234	−0.022	0.132	0.133	−0.169	−0.275	0.084
1-Butanol (X_9_)	0.233	0.12	−0.238	0.018	0.321	−0.06	0.04
1-Pentanol (X_10_)	0.226	0.206	−0.203	−0.029	0.253	−0.072	0.067
1-Propanol (X_11_)	0.215	0.049	−0.108	0.018	0.06	0.078	0.297
Isobutanol (X_12_)	0.212	−0.197	−0.05	−0.345	−0.085	0.001	0.181
Ethyl caprylate (X_13_)	0.202	0.083	−0.007	0.096	0.131	−0.154	0.168
Isoamyl acetate (X_14_)	0.202	−0.195	0.041	−0.126	−0.09	0.273	−0.032
3-Methyl-1-butanol (X_15_)	0.19	−0.172	−0.261	−0.079	0.188	0.087	−0.066
Hexanoic acid (X_16_)	0.086	0.399	−0.063	−0.059	−0.229	−0.009	−0.066
Butyric acid (X_17_)	−0.006	0.314	0.14	0.121	−0.001	−0.174	0.368
Heptanoic acid (X_18_)	0.123	0.293	0.051	−0.03	−0.267	0.186	−0.354
Acetic acid glacial (X_19_)	−0.052	0.283	−0.021	−0.227	−0.313	0.12	0.376
Ethyl phenylacetate (X_20_)	−0.044	0.096	0.445	0.088	0.154	0.03	0.082
Ethyl myristate (X_21_)	0.087	0.023	0.43	0.03	0.103	0.077	−0.383
Hexyl butyrate (X_22_)	−0.022	0.028	0.382	0.202	0.293	−0.04	0.312
1-Hexanol (X_23_)	0.137	0.261	−0.309	0.04	0.252	−0.063	−0.145
Ethyl palmitate (X_24_)	0.119	−0.095	−0.142	0.584	0.054	0.28	−0.011
Ethyl linoleate (X_25_)	0.161	−0.169	−0.066	0.431	−0.226	0.266	0.154
Octanoic acid (X_26_)	0.068	0.227	−0.1	0.284	−0.424	−0.16	−0.071
Ethyl L(-)-lactate (X_27_)	−0.09	0.236	0.038	−0.2	0.203	0.514	0.094
Propionic acid (X_28_)	−0.06	0.175	0.017	0.073	0.007	0.511	0.095

**Table 9 foods-12-02579-t009:** Composition matrix of 28 volatile compounds in strongly flavored Baijiu based on principal component analysis.

Samples	Score
Strong flavor Baijiu made from the upper grains of the mud-sealing pit	0.624 ± 0.117 ^b^
Strong flavor Baijiu made from the middle grains of the mud-sealing pit	
Strong flavor Baijiu made from the lower grains of the mud-sealing pit	
Strong flavor Baijiu made from the upper grains of the steel-sealing pit	0.632 ± 0.052 ^a^
Strong flavor Baijiu made from the middle grains of the steel-sealing pit	
Strong flavor Baijiu made from the lower grains of the steel-sealing pit	

Values are means ± SD. The different letters ^(ab)^ in the same column indicate the values are significantly different (*p* ≤ 0.05).

**Table 10 foods-12-02579-t010:** Predicted results of the typical discriminant analysis of the strongly flavored Baijiu of the upper layer of grains for the two methods for sealing the pits.

	Samples	Results of Prediction						
		MSP-U1^st^	SSP-U1^st^	MSP-U2^nd^	SSP-U2^nd^	MSP-U3^rd^	SSP-U3^rd^	Total
Original count	MSP-U1^st^	21						21
	SSP-U1^st^		15					15
	MSP-U2^nd^			21				21
	SSP-U2^nd^				15			15
	MSP-U3^rd^					21		21
	SSP-U3^rd^						15	15
Proportion (%)	MSP-U1^st^	100						100
	SSP-U1^st^		100					100
	MSP-U2^nd^			100				100
	SSP-U2^nd^				100			100
	MSP-U3^rd^					100		100
	SSP-U3^rd^						100	100

100.0% of original grouped cases were classified correctly. 1^st^, 2^nd^, 3^rd^ represent for the first, second, and third segment strongly flavored Baijiu, respectively.

**Table 11 foods-12-02579-t011:** Predicted results of the typical discriminant analysis of the strongly flavored Baijiu of the middle layer of grains for the two methods for sealing the pits.

	Samples	Results of Prediction						
		MSP-M1^st^	SSP-M1^st^	MSP-M2^nd^	SSP-M2^nd^	MSP-M3^rd^	SSP-M3^rd^	Total
Original count	MSP-M1^st^	18						18
	SSP-M1^st^		15					15
	MSP-M2^nd^			24				24
	SSP-M2^nd^				15			15
	MSP-M3^rd^					21		21
	SSP-M3^rd^						15	15
Proportion (%)	MSP-M1^st^	100						100
	SSP-M1^st^		100					100
	MSP-M2^nd^			100				100
	SSP-M2^nd^				100			100
	MSP-M3^rd^					100		100
	SSP-M3^rd^						100	100

100.0% of original grouped cases were classified correctly. 1^st^, 2^nd^, 3^rd^ represent for the first, second, and third segment strong flavor Baijiu, respectively.

**Table 12 foods-12-02579-t012:** Predicted results of the typical discriminant analysis of the strongly flavored Baijiu of the lower layer of grains for the two methods of sealing the pits.

	Samples	Results of Prediction						
		MSP-L1^st^	SSP-L1^st^	MSP-L2^nd^	SSP-L2^nd^	MSP-L3^rd^	SSP-L3^rd^	Total
Original count	MSP-L1^st^	21						21
	SSP-L1^st^		15					15
	MSP-L2^nd^			21				21
	SSP-L2^nd^				15			15
	MSP-L3^rd^					15		12
	SSP-L3^rd^						21	21
Proportion (%)	MSP-L1^st^	100						100
	SSP-L1^st^		100					100
	MSP-L2^nd^			100				100
	SSP-L2^nd^				100			100
	MSP-L3^rd^					100		100
	SSP-L3^rd^						100	100

100.0% of original grouped cases were classified correctly. 1^st^, 2^nd^, 3^rd^ represent for the first, second, and third segment strong flavor Baijiu, respectively.

## Data Availability

Data is contained within the article.

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
