# Peer review of "Comparative Analysis of Volatile Flavor Compounds in Strongly Flavored Baijiu under Two Different Pit Cap Sealing Processes"

_foods, 2023, doi:10.3390/foods12132579_

Round 1
Reviewer 1 Report
the revision is attached in a separate document.

Author Response
We have added some discussions in the introduction.
- The sealing technology of pit is one of the key points in the production of Chinese Baijiu possessing variedand diversified flavors (12 types including Maotai flavor, strong flavor, Qingxiang flavor, ). The key fermentation elements concerned such as the types of fermentation microorganisms and the corresponding fermentation time are all related to sealing of pit. Poor sealing technology can lead to a series of devastating consequences such as fermentation rancidity, heavy odor, and low alcohol production rate. However, a better sealing technology can make the aroma of the Baijiu stronger and more unique. Please refer to the text for all modifications( see Line64 -Line67) .
- The fermentation of Baijiu is the result of the joint action of microbial clusters. Good fermentation technology will lead to the production of a large number of flavor substances and metabolites, which not only have rich nutritional value but also have certain medicinal value, such as cyclic dipeptide, prochemic acid, etc. . Please refer to the text for all modifications( see Line92 -Line99)
In fact, the main purpose of this ms is to obtain the experimental data support for the reliability of large-scale use of stainless steel sealing technology by comparing the differences or similarities between the stainless steel technology sealing method and the traditional mud cellar sealing method, so that Baijiu brewing can move forward to a greater step in the direction of mechanization and intelligence.

Reviewer 2 Report
The manuscript entitled "Comparative analysis of volatile flavor compounds in strong flavor Baijiu under two different pit cap sealing processes" describes the comparison of sealing process.
In my opinion the manuscript is well organized and fundamented with data from literature. The results are also well discussed.
Comments:
- Abbreviations should be described when used for the first time
- The figures legends should be increased so they can be readable
- Did the authors confirm the identification of compounds? How?
- Table 1, please correct the compounds names for IUPAC. Some names are incorrect
- Tables content fonts should be corrected once they have different numbers
Author Response
- Abbreviations should be described when used for the first time
We corrected
- The figures legends should be increased so they can be readable
We corrected the figures legends and marked them with red colour in the text.
- Did the authors confirm the identification of compounds? How?
Yes, we have determined that the content of the measured compound is within the normal range by consulting relevant literature, and can provide mutual support through the section 3.4, 3.5, and 3.6.
- Table 1, please correct the compounds names for IUPAC. Some names are incorrect
We corrected all similar mistakes in the Table 1.
- Tables content fonts should be corrected once they have different numbers
We revised Table 3.

Reviewer 3 Report
Please find my comments and suggestion regarding the manuscript titled:
"Comparative analysis of volatile flavor compounds in strong flavor Baijiu under two different pit cap sealing processes" - foods-2408911
Results are presented in a good and scientific way.
Up to date literature is used.
My specific comment refer to the following:
English could be checked.
Abbreviations should be written in full names when mentioned for the first time.
A brief description of strong flavor Baijiu should be given in the Introduction.
Figures 1-3 should be placed in the text after they have been mentioned for the first time.
Line 145: N-propanol correct to n-propanol
n-butanol .....
For all compounds with the prefix n- and tert- in the chemical name, n- and tert- should be written in Italic.
The evolution of aroma compounds and their significance should be also described briefly.
Refrences are missing within the text.
Figures 4-9 should be provided in better resolution in order to be more visible
Lines 412-443: The statistical calculation and equations are more suitable for Material and methods section
Section 3.5. should be re-written
Conclusions
Lines 540-543: These lines should be deleted as health related properties of volatile aroma compounds are not discussed in the papaer, nor thez were a subject of the present work.
Conclusion must be re-written including the most relevant findings of the work.
Additional comments are included in the PDF file.
Sincerely,

Moderate editing of English language required
Author Response
English could be checked.
We asked Thomas A. Gavin who is the Professor Emeritus, Cornell University, for help with correcting English in this ms again.
Abbreviations should be written in full names when mentioned for the first time.
We added
A brief description of strong flavor Baijiu should be given in the Introduction.
We added the description of strong flavor Baijiu into the introduction(please see Line 68-Line 71 in revision ms).
Figures 1-3 should be placed in the text after they have been mentioned for the first time.
We revised
Line 145: N-propanol correct to n-propanol
n-butanol .....
For all compounds with the prefix n- and tert- in the chemical name, n- and tert- should be written in Italic.
We revised
The evolution of aroma compounds and their significance should be also described briefly.
We added the evolution of aroma compounds and their significance into the introduction(please see Line 71-Line 73 in revision ms).
Refrences are missing within the text.
Actually, there are references in the discussion section of the text. Most of it is experimental data.
Figures 4-9 should be provided in better resolution in order to be more visible
We redrew Figures 4-9, now they could be more clear than before.
Lines 412-443: The statistical calculation and equations are more suitable for Material and methods section
In fact, Lines 412-443 is the calculated data result, and Table 9 is calculated based on this result.
Section 3.5. should be re-written
We have deleted some of the content.
Conclusions
Lines 540-543: These lines should be deleted as health related properties of volatile aroma compounds are not discussed in the papaer, nor thez were a subject of the present work.
We deleted "The volatile compounds with antibacterial, analgesic, and anti-inflammatory activity were found in the Baijiu brewed by using the steel sealing pit, which provided a direction for subsequent in-depth research on the beneficial compounds, such as in-liquor compo-nents of Chinese Baijiu (i.e., health functional factors). "
Conclusion must be re-written including the most relevant findings of the work.
We rewrote this paragraph, now it should include the most relevant findings of the work, hopefully.
Additional comments are included in the PDF file.
We deleted some unnecessary comments in the PDF file.

Round 2
Reviewer 1 Report
The manuscript was partially revised and supplemented in Introduction part. However, the hypothesis is not sufficiently explained. The topic of the manuscript is not relevant and not suited to Food journal (Q1). The manuscript is devoted on improvement technology of Baijiu alcoholic beverage thus may significance on the local market.
Author Response
As the statement of the reviewer, this MS is committed to improving the production process of alcoholic beverages. The annual output value of Baijiu in China alone in Sichuan Province is more than trillion RMB. The mechanized and intelligent upgrading and improvement of the process can significantly improve the productivity and efficiency of Baijiu enterprises, and even double and triple the output value.It has great theoretical guidance for the development of mechanization and intelligence in future Baijiu(liquor) enterprises.
As for the hypothesis, another reviewer also asked, is the MS focusing on quality or quantity? We explained “We are focusing on the Quality. Based on the almost constant quality of the two sealing methods, make the mechanization and intelligence of brewing becoming possible, ultimately achieving a significant increase in Quantity of Baijiu (means an increase in economic benefits).”

Reviewer 3 Report
Although authors made some corrections in the paper, still there are some issues that should be resolved.
Lines 41-48: Are these lines belong to the Abstract?
If so, corrections should be made, and text should be added to the Abstract lines.
Lines 64-66 are added by authors:
“ The fermentation of Baijiu is the result of the joint action of microbial clusters. Good fermentation technology will lead to the production of a large number of flavor substances and metabolites, which not only have rich nutritional value but also have certain medicinal value, such as cyclic dipeptide, prochemic acid, etc[5-6].”
These lines do not describe the formation of aromatic compounds in the studied beverage. In addition, the references added by authors are not much relevant to the issue raised (evolution and significance of aroma compounds).
5. Fan, M.; Yuan, S.; Li, L.; Zheng, J.; Zhao, D.; Wang, C.; Wang, H.; Liu, X.; Liu, J. Application of Terpenoid Compounds in Food and Pharmaceutical Products. Fermentation 2023, 9, 119. https://doi.org/10.3390/fermentation9020119 616
6. Yuan et al.,2020. Research Progress of the Biosynthesis of Natural Bio-Antibacterial Agent Pulcherriminic Acid in Bacillus. 617 Molecules, 25: 5611.
It should be corrected as previously suggested. A paper of Matijašević et al. Molecules, 2019, 24, 2485; https://www.mdpi.com/1420-3049/24/13/2485, and references cited in there might be useful.
Refrences are still missing in some places; for aroma desriptors stil no reference (or references) is included.
Lines 288-291: In addition, the following volatile compounds with antibacterial, analgesic, and an-ti-inflammatory activity were found in the middle fermented layer of grains in the steel-sealing pit: L-pyroglutamic acid methyl ester, and Cyclo (Phe-Pro). These compounds are beneficial when consumed and provide a basis for the direction of subsequent research on beneficial flavor compounds in this brand of Chinese Baijiu.
No refrence is added for the following paragraph, although it has been previously suggested.
Tables 3-5 should be placed in landscape as many figures appear in two rows and make them hard for reading. Moreover, tables in the present form look messy.
In Conclusions: Mutual evidence of 577 HCA, DA, and PCA.
What does it mean? Please explain.
At the end, what was the focus of the study? Quality or quantity?
I am not sure what authors meant by: „We deleted some unnecessary comments in the PDF file.“
Additional comments were previously given to the authors in order to point out some text that should be revised.
Author Response
Reviewer3#:
1.Lines 41-48: Are these lines belong to the Abstract?
Yes, it belongs to the Abstract.
2.If so, corrections should be made, and text should be added to the Abstract lines.
Sorry, we didn't understand the meaning of the reviewer. What text should we add to the Abstract lines?
3.Lines 64-66 are added by authors:
“ The fermentation of Baijiu is the result of the joint action of microbial clusters. Good fermentation technology will lead to the production of a large number of flavor substances and metabolites, which not only have rich nutritional value but also have certain medicinal value, such as cyclic dipeptide, prochemic acid, etc[5-6].”
These lines do not describe the formation of aromatic compounds in the studied beverage. In addition, the references added by authors are not much relevant to the issue raised (evolution and significance of aroma compounds).
- Fan, M.; Yuan, S.; Li, L.; Zheng, J.; Zhao, D.; Wang, C.; Wang, H.; Liu, X.; Liu, J. Application of Terpenoid Compounds in Food and Pharmaceutical Products. Fermentation 2023, 9, 119. https://doi.org/10.3390/fermentation9020119 616
- Yuan et al.,2020. Research Progress of the Biosynthesis of Natural Bio-Antibacterial Agent Pulcherriminic Acid in Bacillus. 617 Molecules, 25: 5611.
It should be corrected as previously suggested. A paper of Matijašević et al. Molecules, 2019, 24, 2485; https://www.mdpi.com/1420-3049/24/13/2485, and references cited in there might be useful.
We added the formation of aromatic compounds into the introduction(please see Line75-Line86 in revision ms).
In fact, this paragraph “The fermentation of Baijiu is the result of the joint action of microbial clusters. Good fermentation technology will lead to the production of a large number of flavor substances and metabolites, which not only have rich nutritional value but also have certain medicinal value, such as cyclic dipeptide, prochemic acid, etc[5-6].”It was requested to be supplemented by another reviewer. We have already replaced references
4.Refrences are still missing in some places; for aroma desriptors stil no reference (or references) is included.
We added(reference [8-12], reference [35]).
5.Lines 288-291: In addition, the following volatile compounds with antibacterial, analgesic, and an-ti-inflammatory activity were found in the middle fermented layer of grains in the steel-sealing pit: L-pyroglutamic acid methyl ester, and Cyclo (Phe-Pro). These compounds are beneficial when consumed and provide a basis for the direction of subsequent research on beneficial flavor compounds in this brand of Chinese Baijiu.
No refrence is added for the following paragraph, although it has been previously suggested.
We added(Line298).
6.Tables 3-5 should be placed in landscape as many figures appear in two rows and make them hard for reading. Moreover, tables in the present form look messy.
We revised, please see the revision of Tables 3-5.
7.In Conclusions: Mutual evidence of 577 HCA, DA, and PCA. What does it mean? Please explain.
HCA can divide samples into two categories, and using DA can also separate samples without aggregation. At the same time, two types of samples (one is mud-sealing and the other is steel-sealing) were successfully identified. At the first, the relationship between the two methods is mutually verified, by this way it can be resulting in a more robust conclusion.(Line551--Line555).
8.At the end, what was the focus of the study? Quality or quantity?
Quality.
Based on the almost constant quality of the two sealing methods, make the mechanization and intelligence of brewing becoming possible, ultimately achieving a significant increase in quantity of Baijiu (means an increase in economic benefits).
9.I am not sure what authors meant by: “We deleted some unnecessary comments in the PDF file. ”
Our original intention was to remove some reviewer’s comments(like below picture shown lines 180-182) , and finally, it was found that this response was unnecessary.
